# An Overview of Privacy Dimensions on the Industrial Internet of Things (IIoT)

**Vasiliki Demertzi [1], Stavros Demertzis [2] and Konstantinos Demertzis [3,*]**

[1] Computer Science Department, School of Science, Kavala Campus, International Hellenic University, 65404 Kavala, Greece

[2] Faculty of Engineering, School of Spatial Planning and Development, Aristotle University of Thessaloniki, 54124 Thessaloniki, Greece

[3] School of Science & Technology, Informatics Studies, Hellenic Open University, 26335 Patra, Greece

* Correspondence: demertzis.konstantinos@ac.eap.gr

**Abstract:** The rapid advancements in technology have given rise to groundbreaking solutions and practical applications in the field of the Industrial Internet of Things (IIoT). These advancements have had a profound impact on the structures of numerous industrial organizations. The IIoT, a seamless integration of the physical and digital realms with minimal human intervention, has ushered in radical changes in the economy and modern business practices. At the heart of the IIoT lies its ability to gather and analyze vast volumes of data, which is then harnessed by artificial intelligence systems to perform intelligent tasks such as optimizing networked units' performance, identifying and correcting errors, and implementing proactive maintenance measures. However, implementing IIoT systems is fraught with difficulties, notably in terms of security and privacy. IIoT implementations are susceptible to sophisticated security attacks at various levels of networking and communication architecture. The complex and often heterogeneous nature of these systems makes it difficult to ensure availability, confidentiality, and integrity, raising concerns about mistrust in network operations, privacy breaches, and potential loss of critical, personal, and sensitive information of the network's end-users. To address these issues, this study aims to investigate the privacy requirements of an IIoT ecosystem as outlined by industry standards. It provides a comprehensive overview of the IIoT, its advantages, disadvantages, challenges, and the imperative need for industrial privacy. The research methodology encompasses a thorough literature review to gather existing knowledge and insights on the subject. Additionally, it explores how the IIoT is transforming the manufacturing industry and enhancing industrial processes, incorporating case studies and real-world examples to illustrate its practical applications and impact. Also, the research endeavors to offer actionable recommendations on implementing privacy-enhancing measures and establishing a secure IIoT ecosystem.

**Keywords:** identity privacy; location privacy; footprint privacy; multidimensional privacy; privacy threats; privacy principles



## 1. Introduction

Conventional corporate structures, within a society where humans and robots need to collaborate, create obstacles that squander energy, devalue information, and limit knowledge transfer. At this point, the IIoT is transforming how businesses and, by extension, the manufacturing industry, operate [1]. The IIoT is relevant to ubiquitous connectivity, enabling objects, machines, and devices to exchange data autonomously across networks, eliminating the need for human intervention. This connectivity has brought about remarkable progress in industrial organizations, seamlessly merging the physical and digital realms and driving transformative changes in the economy and modern business practices. With the aid of sensors, businesses can effectively monitor performance and address areas that require improvement, thereby enhancing both the manufacturing process and the overall customer experience and ultimately adding value at every production stage [2].

In this context, the IIoT in industrial production is one of the most forward-thinking industrial technologies because it combines two digitization strategies, Artificial Intelligence (AI) and Industry 4.0 [3]. The combination of data analysis and high-level automation technologies significantly improves the industrial environment [4].

However, implementing IIoT systems is fraught with difficulties, notably in terms of security and privacy. The complex and heterogeneous nature of IIoT systems makes them vulnerable to sophisticated security attacks at various levels of networking and communication architecture. Ensuring availability, confidentiality, and integrity becomes a daunting task due to the chaotic nature of these systems. Consequently, there is a risk of mistrust in network operations, concerns about privacy breaches, and the potential loss of critical, personal, and sensitive data of the network's end-users.

The comparison outlining the advantages and disadvantages of IIoT devices is presented in Table 1.

**Table 1.** Advantages and disadvantages of IIoT devices.

| Advantages | |
|---|---|
| Enhanced efficiency and productivity | IIoT devices can help to streamline and automate industrial processes, reducing manual labor and increasing efficiency and productivity. |
| Improved data collection and analysis | IIoT devices can generate large amounts of data that can be used to monitor and optimize industrial processes, leading to better decision-making and improved outcomes. |
| Cost savings | IIoT devices can help to reduce costs by optimizing energy usage, reducing waste, and improving asset management. |
| Remote monitoring and control | IIoT devices can be monitored and controlled remotely, reducing the need for on-site personnel and enabling real-time monitoring and intervention. |
| **Disadvantages** | |
| Security risks | IIoT devices are vulnerable to cyberattacks and hacking, compromising sensitive data and disrupting industrial processes. |
| Compatibility issues | IIoT devices may not be compatible with existing industrial systems or other IIoT devices, leading to integration challenges and additional costs. |
| Lack of standardization | There currently needs to be a widely accepted standard for IIoT devices, leading to issues with interoperability and compatibility. |
| High implementation costs | IIoT devices can be expensive to implement and maintain, particularly for small and medium-sized enterprises. |

One of the biggest challenges is the industrial privacy requirements in an IIoT ecosystem. Industrial privacy refers to the protection of sensitive information and personal data in the context of industrial environments, such as manufacturing plants, supply chains, and industrial processes. It encompasses the safeguarding of confidential business information, trade secrets, employee data, and customer information from unauthorized access, misuse, or disclosure. Industrial privacy is crucial for maintaining the confidentiality, integrity, and availability of sensitive data within the industrial sector. It involves implementing security measures, policies, and practices that address the unique challenges and risks associated with industrial settings, including the interconnectedness of industrial systems, the collection and analysis of machine-generated data, and the integration of new IIoT technologies and applications. Industrial privacy aims to ensure that sensitive information remains protected, both within the organization and in its interactions with external stakeholders,

contributing to maintaining trust, compliance with regulations, and the sustainable growth of industrial sectors.

This research endeavors to address several key aspects concerning the privacy requirements within an IIoT ecosystem, particularly in relation to the processing of personal data by competent authorities. Specifically, the research questions addressed in this study are:

What are the specific industrial privacy requirements within an IIoT ecosystem, particularly in relation to the processing of personal data by competent authorities?

1. What are the existing approaches and solutions used to mitigate privacy risks in industrial settings?
2. How can privacy dimensions be identified and defined to safeguard individuals within the IIoT ecosystem? What are the different aspects or components of privacy that should be considered?
3. What are the contemporary techniques, technologies, and best practices employed to ensure data privacy and security in IIoT systems within industrial environments?
4. How can organizations establish an ideal, safe, and private IIoT ecosystem within the industrial domain?
5. What considerations and factors need to be taken into account to create a secure environment within the industrial domain while adhering to relevant industry standards?
6. What are the recommendations for implementing privacy-enhancing measures in IIoT systems to effectively manage privacy risks and ensure compliance with relevant regulations?

By addressing these research questions, the study aims to provide insights into the challenges and strategies for maintaining industrial privacy within the context of IIoT ecosystems. It also seeks to offer guidance to organizations in enhancing data protection measures, thereby fostering trust, compliance, and sustainable growth within the industrial sectors. In summary, this study makes the following significant contributions to understanding and implementing the IIoT in the industrial context:

1. IIoT's Transformative Impact on Industrial Organizations: The introduction emphasizes how the Industrial Internet of Things (IIoT) is revolutionizing the manufacturing industry by facilitating ubiquitous connectivity and autonomous data exchange. This transformation is breaking down barriers between physical and digital realms, leading to remarkable progress in industrial operations and modern business practices.
2. Integration of AI and Industry 4.0 in the IIoT: The paper highlights that the IIoT represents a forward-thinking industrial technology that combines two digitization strategies—Artificial Intelligence (AI) and Industry 4.0. This integration of data analysis and high-level automation significantly enhances the industrial environment and operations.
3. Challenges in IIoT Implementation: The introduction acknowledges the challenges associated with the IIoT, particularly in terms of security. The complex and heterogeneous nature of IIoT systems makes them vulnerable to sophisticated security attacks at various levels of networking and communication architecture. These challenges can lead to mistrust in network operations, privacy breaches, and the loss of critical data.
4. Advantages and Disadvantages of IIoT Devices: The introduction presents a clear and concise list outlining the advantages and disadvantages of IIoT devices in Table 1. The benefits include enhanced efficiency, improved data collection, cost savings, and remote monitoring. On the other hand, the disadvantages include security risks, compatibility issues, lack of standardization, and high implementation costs.
5. Focus on Industrial Privacy in IIoT Ecosystems: The research aims to address the crucial aspect of industrial privacy in an IIoT ecosystem. It defines industrial privacy as the protection of sensitive information and personal data within industrial settings, such as manufacturing plants and supply chains. The research emphasizes the need to implement security measures, policies, and practices to safeguard confidential information from unauthorized access and misuse.

6. Identification of Privacy Dimensions: The study delves into the identification and definition of privacy dimensions within the IIoT ecosystem. These dimensions represent different aspects or components of privacy that are considered when addressing privacy concerns. By understanding the multifaceted nature of privacy, the research aims to guide the development of privacy frameworks and practices.

7. Techniques and Best Practices for Data Privacy and Security: The paper explores contemporary techniques, technologies, and best practices used to ensure data privacy and security in IIoT systems. It includes an analysis of the latest methodologies and tools implemented to maintain data confidentiality, integrity, and availability in industrial environments.

8. Establishing a Safe and Private IIoT Ecosystem: The research aims to identify how organizations can establish an ideal, safe, and private IIoT ecosystem within the industrial domain. It delves into various considerations and factors that need to be taken into account to create a secure environment. Additionally, the study offers recommendations on implementing privacy-enhancing measures and adhering to industry standards to effectively manage privacy risks and regulatory compliance.

## 2. Related Work

The complexity and heterogeneity of IIoT systems make ensuring availability, confidentiality, and integrity difficult. As a result, there is a risk of mistrust in network operations, privacy concerns, and the potential loss of critical personal and sensitive information of the network's end-users. The above challenges have caused widespread concern in the scientific community, prompting several solutions to be proposed at various levels. For example, the existing methods for protecting location privacy are mostly based on traditional anonymization, fuzzy, and cryptography technology, with little success in the big data environment, for example, sensor networks contain sensitive information that must be appropriately protected. Current trends, such as "Industry 4.0" and the IIoT, generate, process, and exchange massive amounts of security-critical and privacy-sensitive data, making them appealing targets for cyber-attacks. However, the previous methods ignored the issue of privacy protection, resulting in a violation of privacy. In this paper [5], the authors propose a location privacy protection method that meets the differential privacy constraint while also maximizing the utility of the data and algorithms in the IIoT. Since location data has a high value but a low density, the authors combine utility and privacy to create a multilevel location information tree model. Furthermore, the differential privacy index mechanism is used to select data based on the frequency with which tree nodes access the data. Finally, the Laplace scheme is used to introduce noise into the data accessing frequency. As demonstrated by the theoretical analysis and experimental results, the proposed strategy can significantly improve security, privacy, and applicability.

A tensor-based multiple clustering method has recently been created with the goal of uncovering hidden distinct data structures in large data from diverse perspectives, and it may be widely employed in the IIoT to improve production and service quality. Yet, due to the high computational cost and massive volume of data, outsourcing processing to relatively low-cost cloud servers can significantly reduce local costs, but there is a considerable risk of revealing user privacy. To overcome the aforementioned issue, a secure hybrid cloud-based tensor-based multiple clustering method is proposed [6]. The proposed technique encrypts object tensors using a homomorphic cryptosystem and then uses cloud servers to completely conduct various clustering calculations over encrypted object tensors. A set of related security subprotocols is also developed to facilitate privacy-preserving tensor-based multiple clusterings. Just encryption and perturbation removal are performed on the client in the proposed method, making it very lightweight for consumers. Experiment findings show that the suggested approach is accurate and efficient when grouping items into different groups, with no leakage of private or supplementary information. Furthermore, when more cloud nodes are used, the technique offers excellent scalability, making it ideal for clustering Industrial IIoT big data.

In addition, clients that are equipped with growing cloud computing choose to outsource an increasing amount of IIoT data to the cloud to alleviate the high storage and processing strain. Existing searchable encryption (SE) systems, however, only apply to IIoT records with textual keyword fields, rather than those with both digital and textual keyword fields. Furthermore, due to the significant key storage expense, the key management issue continues to limit the practicality and availability of SE schemes. To that purpose, the authors [7] describe an outsourced Hybrid Keyword-Field Search over Encrypted Data with Efficient Key Management (HKFS-KM) system that makes use of the relevance score function and a keyed hash tree. A formal security study demonstrates that the HKFS-KM scheme achieves keyword privacy and trapdoor unlinkability in both known ciphertexts and known background attack models. The experimental findings utilizing real-world datasets demonstrate its efficiency and applicability in practice.

Although cyber-attacks on the Industrial Internet of Things (IIoT) continue to be a serious concern, blockchain has emerged as a viable technology for IIoT security due to its decentralization and immutability. Current blockchain designs, on the other hand, provide considerable computational complexity and latency concerns, making them unsuitable for the IIoT. Xyreum, a novel high-performance and scalable blockchain for increased IIoT security and privacy, is proposed in this research [8]. To accomplish Mutual Multi-Factor Authentication, Xyreum employs a Time-based Zero-Knowledge Proof of Knowledge (T-ZKPK) with authenticated encryption (MMFA). T-ZKPK characteristics are also utilized to help with Key Establishment for transaction security. Their approach to establishing consensus, which is a group decision-making process on the blockchain, is based on lightweight cryptographic techniques. They test their scheme for security, privacy, and performance, and the results show that, when compared to existing relevant blockchain solutions, their scheme is secure, privacy-preserving, and achieves a significant reduction in computation complexity and latency performance while maintaining high scalability. They also demonstrate a blockchain-based security protocol used in a variety of application domains.

Moreover, deep learning offers a great chance to extract usable knowledge from the massive amounts of data in the IIoT. Yet, the absence of large public datasets will result in poor performance and overfitting of the learned model. As a result, federated deep learning across distant datasets has been proposed. Yet, it invariably brings new security challenges, such as revealing participant data privacy. Existing solutions, however, cannot ensure the privacy of each participant's data in a learning group. The authors [9] suggest two privacy-preserving asynchronous deep learning systems in this article: DeepPAR (privacy-preserving and asynchronous deep learning via re-encryption) and DeepDPA (dynamic privacy-preserving and asynchronous deep learning). In comparison to previous work, DeepPAR secures each participant's input privacy while maintaining dynamic update secrecy naturally. Meanwhile, DeepDPA allows for the backward privacy of group participants to be guaranteed in a lightweight manner. Security analyses and performance tests on real-world datasets demonstrate that their suggested systems are safe, efficient, and effective.

On the other hand, the rapid increase in the volume of data created by the connected devices in the IIoT paradigm brings up new opportunities for improving service quality for developing applications through data sharing. Yet, data providers have significant challenges in sharing their data through wireless networks due to security and privacy concerns (e.g., data leakage). Private data leaks can cause major problems beyond financial loss for companies. The authors [10] first design a blockchain-powered safe data-sharing architecture for the multiple distributed parties in this study. Finally, using privacy-protected federated learning, they transform the data-sharing challenge into a machine-learning problem. Data privacy is protected by providing the data model rather than releasing the actual data. Lastly, they incorporate federated learning into the permissioned blockchain consensus process, such that the computing work for the consensus can also be used for federated training. The suggested data-sharing strategy achieves good accuracy, high efficiency, and better security, according to the numerical results generated from the real-world datasets.

Finally, the advantage of using edge computing is that it may be used as a complement to cloud computing; blockchain is an alternative for creating a transparent safe environment for data storage/governance. Instead, rather than employing these two strategies separately, the authors propose a novel methodology in this article [11], termed the blockchain-based Internet of Edge model, which merges the IIoT with edge computing and blockchain. The suggested approach, intended for a scalable and controllable IIoT system, takes advantage of the benefits of edge computing and blockchain to construct a privacy-preserving mechanism while taking into account other constraints such as energy cost. They carry out experiment evaluations on Ethereum. According to their data gathering, the proposed strategy improves privacy safeguards while reducing energy consumption.

In the rapidly evolving landscape of the IIoT, the research community, as mentioned above, has embraced various privacy methods to protect sensitive data and ensure secure operations. However, each method comes with its own set of disadvantages, highlighting the necessity for a holistic approach when addressing privacy dimensions in the IIoT ecosystem. From this point of view, the IIoT ecosystem's privacy dimensions require a comprehensive approach that takes into account the interconnectedness of the various methods and their associated advantages. By presenting a holistic approach to privacy in the IIoT, industries and organizations can strengthen their defenses against evolving threats and protect sensitive data, ensuring the secure and sustainable growth of IIoT technologies.

## 3. IIoT and Privacy-Preserving Architectures

Unlike traditional industrial systems that operate in isolation or with limited connectivity, IIoT systems enable seamless communication between devices and systems, both within a single industrial facility and across multiple sites or locations. This interconnectedness allows for the exchange of data and information, facilitating the integration of operational technology (OT) with information technology (IT) systems [12].

The IIoT leverages technologies such as cloud computing, big data analytics, machine learning, and artificial intelligence to process and derive insights from the vast amount of data generated by industrial devices and systems. These technologies enable predictive maintenance, remote monitoring, intelligent automation, and optimization of industrial processes.

The applications of the IIoT are diverse and can be found in various industries, including manufacturing, energy and utilities, transportation, agriculture, healthcare, and more. Examples of IIoT implementations include smart factories, connected supply chains, remote asset management, predictive maintenance, and energy management systems.

Any edge device (sensors, readers, gateways) can transfer local data to cloud systems by using any available communication system for real-time analysis. However, if they are not incorporating AI applications, their use can be considered passive, as they cannot utilize the data in real-time.

AI plays a crucial role in leveraging data in real-time by enabling efficient processing, analysis, and decision-making. Also, AI empowers organizations to harness the power of real-time data by enabling efficient processing, analysis, and decision-making. By leveraging AI algorithms and techniques, organizations can extract insights, make predictions, automate processes, and provide personalized experiences in real-time, thereby gaining a competitive edge and driving innovation. It must be noted that real-time data processing focuses on the timely processing and analysis of data as it is generated, enabling organizations to make immediate decisions or take prompt actions. Intelligent automation, on the other hand, leverages AI and automation technologies to automate tasks and processes, reducing human effort and improving efficiency. Together, they can enhance operational agility, optimize decision-making, and drive intelligent actions based on real-time data.

On the other hand, this closer networking of the digital world of machines creates the potential for profound changes in global industry and many areas of private and social life. Based on all this, it is necessary to present tomorrow's trends in everything related to IIoT technology applications [13].

1.　Growth of IIoT applications [14]. Manufacturing automation continues to grow, with the number of companies choosing to automate and implement the IIoT soaring to new levels due to the impact of the COVID-19 pandemic. Machine learning and robotics are two applications that increase automation. Machine learning increasingly automates manufacturing processes, so less human intervention is required, while the increasing number of human jobs being taken over by robotics results in fewer people in the workplace. Organizations need to ensure that proper security protocols are in place to safeguard data privacy and prevent unauthorized access.

2.　The wireless revolution. Only some IIoT applications have access to local sockets [15]. "Local socket" refers to a communication endpoint or interface that allows processes running on the same device or within the same local network to communicate with each other. Local sockets in IIoT architectures can provide privacy in industrial environments by enabling secure and private communication between processes and applications running on the same device or within the same local network. By utilizing local sockets, data can be exchanged and coordinated within the confines of the device or local network, reducing the risk of unauthorized access or interception from external sources. This local communication ensures that sensitive data stays within the trusted boundaries of the industrial environment, enhancing privacy and preventing potential security breaches. Additionally, with the advent of advanced IIoT wireless technologies like 5G, organizations can further enhance network isolation and security, creating dedicated and isolated network environments that offer heightened privacy, control, and protection for sensitive data through features such as Network Slicing, Enhanced Security Mechanisms, Private 5G Networks, and Network Function Virtualization (NFV). Secure and private communication between processes and applications within the industrial environment helps maintain data privacy and prevents potential security breaches.

3.　Adoption of Virtual Reality (VR) [16] for remote operations has become dominant for industrial applications regarding training and commissioning. Devices that combine a screen, camera, and microphone have become more sophisticated, and machine suppliers more often collaborate with their customers or service engineers through VR. The ability to commission machines remotely has made companies realize that being on-site is only sometimes necessary. The machine supplier can work with the customer through an augmented reality headset, such as a HoloLens. The customer sees virtual reality instructions and maintenance data to perform the necessary tasks, while the machine supplier receives a live feed of what the customer sees. As companies employ VR for training, maintenance, and collaboration purposes, it is crucial to ensure that privacy safeguards are in place to protect sensitive information shared through these immersive technologies.

4.　Use of machine data to improve customer relations [17,18]. Connected machines have opened new ways to use machine data and improve customer relationships. The above statement highlights the impact of interconnected devices and the data they generate on enhancing customer relationships. Specifically, connected machines and the data they generate enable organizations to leverage the machine data in various ways, leading to improved customer relationships. By utilizing machine data for proactive maintenance, predictive analytics, customized offerings, and enhanced support, organizations can deliver better customer experiences, increase customer satisfaction, and strengthen their relationships with the customers. It is not only interesting for large companies but also for smaller companies to make use of their data. Due to the increase in connected machines, the number of companies with access to critical machine data has also increased tremendously. It is a big challenge for many companies to discover new possibilities. The use of data is not only essential to improve and optimize companies' machines but also to create a better long-term relationship with customers. Machine data can, for example, be used to prevent equipment failures by predicting and performing machine maintenance before a fault

occurs. In this way, machine downtime can ultimately be reduced [19]. Ensuring data security and using anonymization techniques when analyzing and utilizing machine data can help protect customer privacy.

5.  Machine learning [20]. Machine learning is a branch of AI, where systems must be able to learn automatically and improve from experience without being programmed by humans. Applying machine learning can be difficult because preprocessing to label and normalize many data takes time. Unsupervised learning or self-learning methodologies create higher-scale automation [21]. This means that human intervention is no longer needed since the data from the device is automatically sent to the algorithm. Thus, machine learning detects patterns of normal usage; therefore, after some time, it also tracks unusual patterns. For example, a machine creates several terns, but when a part of the machine fails, new patterns are created with donations from the usual pattern. When such a situation occurs, machinery suppliers receive a notification so they know that maintenance is required [22,23]. Implementing data privacy and security measures during data preprocessing, model training, and inference stages is crucial to maintaining privacy while benefiting from machine learning techniques.

6.  "Smart" packaging [24]. Using direct materials with built-in connections, intelligent packaging delivers advanced benefits for industries. A fundamental feature of smart packaging is enabling customers to interact with it and collect data for more efficient product handling. Smart packaging may include video recipes and other demonstrations that explain the product's use. ICT and packaging interact in several ways, including sensors, Quick Response (Q.R.) codes, and augmented/virtual/mixed reality possibilities. The objective is to increase the customer value and data collection via intelligent monitoring to optimize the operations and improve efficiency [25]. Ensuring transparent data collection practices, obtaining informed consent, and implementing robust security measures helps to protect customer privacy and build trust.

As can be easily seen, the development of the IIoT is a big step in realizing Industry 4.0 and the upcoming Industry 5.0, as it promotes the large-scale automation and optimization of processes related to intelligent sensors (e.g., configuration, high-volume handling of data, decision-making, etc.). But this involves significant technical difficulties due to the industrial wireless networks' large scale and complex structure. In addition, recording and transmitting large amounts of data creates severe security and privacy concerns, as some may contain sensitive industry and personal information [26].

Privacy and security in the IIoT scenario are presented in Figure 1 [27].

The ADVOCATE approach aims to address data privacy and consent management in various user environments, such as smart homes, patient health monitoring systems, and activity monitoring sensors. It utilizes a portable device, like a mobile phone, to create a user-friendly interface for data subjects to interact with and manage their personal data disposal policy and consent.

The architecture proposed by ADVOCATE focuses on three specific ecosystems: smart cities, industry, and healthcare. These environments often involve a wide range of sensors and devices that collect data about individuals. By using the ADVOCATE approach, data controllers can interact with data subjects through the portable device to obtain their consent for the data processing activities.

In addition, in the industrial sector, the ADVOCATE approach is applied to ensure that data subjects have a say in how their personal data is processed within industrial environments. This is particularly important considering the sensitive nature of data involved in manufacturing processes, trade secrets, and industrial control systems. By using a portable device, individuals can easily manage their consent preferences and ensure that their personal data is handled appropriately.

It must be noted that ADVOCATE is an ideal industry privacy paradigm by providing a user-centric and customizable framework for managing consent, data disposal policies, and privacy preferences. It helps industries comply with privacy regulations, address

industry-specific privacy concerns, and empower individuals to have control over their personal data in industrial environments.

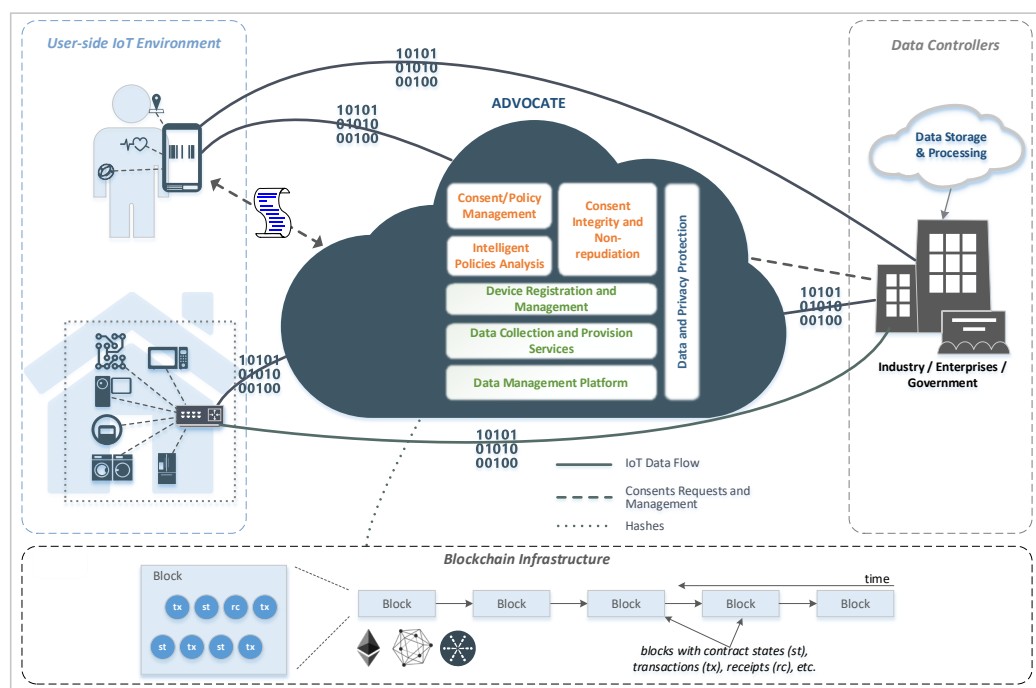

**Figure 1.** Privacy and security in the IoT/IIoT using ADVOCATE architecture [27].

Architectures that protect privacy are promising solutions to the IIoT ecosystem. Privacy-preserving architectures refer to the design and implementation of systems that prioritize the protection and preservation of user privacy. These architectures are particularly relevant in today's digital age, where vast amounts of personal data are collected, processed, and shared. The following are some commonly employed privacy-preserving architectures [26]:

1.  Privacy by Design (PbD) [28]: Privacy by Design is a framework that promotes privacy considerations throughout the entire system development lifecycle. It involves embedding privacy features and measures into the architecture, ensuring that privacy is a core principle from the initial design stages. PbD can certainly be applied in IIoT environments. By integrating privacy considerations into the design and development of IIoT systems, organizations can ensure that privacy is a fundamental aspect of their architecture and processes.

2.  Differential Privacy [29]: Differential privacy is a technique that aims to protect individual privacy while still allowing useful information to be extracted from datasets. It adds noise or perturbation to the data to prevent the identification of specific individuals while preserving the overall statistical properties of the dataset. Differential privacy can be challenging to implement in IIoT environments due to the decentralized and diverse nature of the data sources. However, with careful design and data aggregation techniques, it is possible to apply differential privacy principles in certain IIoT use cases where data privacy is crucial.

3.  Federated Learning [30]: Federated learning is an approach where machine learning models are trained on decentralized data without transferring the data to a central server. This architecture allows for collaborative model training while keeping the data on individual devices, thereby maintaining privacy. Federated learning can be well-suited for IIoT environments, as it allows collaborative model training while keeping the data on individual devices or local servers. This approach preserves privacy by minimizing data transfer and centralization.

4. Homomorphic Encryption [31]: Homomorphic encryption enables computation on encrypted data without decrypting it. It allows data to be processed securely in an encrypted state, preserving privacy during computation. Homomorphic encryption can be complex to implement in resource-constrained IIoT devices and systems due to its computational overhead. However, advancements in hardware and cryptographic techniques may make it more feasible for specific IIoT use cases where privacy-preserving computations are necessary.

5. Zero-Knowledge Proofs [32]: Zero-knowledge proofs are cryptographic protocols that allow one party to prove the validity of certain information to another party without revealing the actual information. This approach enables the verification of data or statements without exposing the underlying sensitive data. Zero-knowledge proofs can be challenging to implement in IIoT environments due to the limited computational capabilities and communication constraints of IIoT devices. However, they can be applied in certain scenarios where privacy-preserving authentication or verification is required.

6. Data Minimization [33]: Data minimization involves collecting and retaining only the necessary data for a specific purpose, reducing the exposure of personal information. By limiting the amount of data collected, processed, and stored, privacy risks are reduced. Data minimization is highly relevant and applicable in IIoT environments. Limiting the collection, processing, and retention of personal data to what is strictly necessary helps reduce privacy risks and ensures compliance with privacy regulations.

7. User-centric Identity and Access Management (IAM) [34]: User-centric IAM puts individuals in control of their personal information. It allows users to manage their own identity and control the sharing of their personal data, ensuring privacy preferences are respected. User-centric IAM may have limited applicability in IIoT environments since the concept of individual users may not always align with the industrial setting. However, similar principles can be applied to manage access, authentication, and authorization of IIoT devices and systems, ensuring that privacy preferences are respected.

These privacy-preserving architectures aim to strike a balance between the need to collect and process data for functional purposes while respecting individual privacy rights. By incorporating privacy-enhancing technologies and principles, these architectures help mitigate privacy risks and build trust between users and service providers.

Blockchain technology offers several privacy-preserving architectures that aim to protect sensitive data while leveraging the benefits of a decentralized and immutable ledger. Here are some key privacy-preserving architectures in blockchain [35]:

1. Confidential Transactions [36]: Confidential transactions use cryptographic techniques to obfuscate transaction details while still maintaining the integrity of the blockchain. This allows for the concealment of transaction amounts and participant identities, enhancing privacy.

2. Zero-Knowledge Proofs [37]: Zero-knowledge proofs (ZKPs) enable the verification of certain statements or computations without revealing the underlying data. ZKPs can be utilized in blockchain to prove the validity of transactions or smart contract conditions without disclosing the sensitive information involved.

3. Ring Signatures [38]: Ring signatures allow for the anonymous signing of a transaction on behalf of a group. In a blockchain context, a ring signature enables a participant to sign a transaction without revealing their specific identity, making it difficult to determine the actual signer.

4. Stealth Addresses [39]: Stealth addresses provide privacy in transactions by creating a one-time destination address for each transaction. This prevents the direct association between a sender's address and the recipient's address, enhancing privacy.

5. Homomorphic Encryption [40]: Homomorphic encryption enables computations to be performed on encrypted data without decrypting it. By applying this technique

to blockchain, sensitive data can be stored and processed in an encrypted state, preserving privacy.

6.  Zero-Knowledge Succinct Non-Interactive Arguments of Knowledge (zk-SNARKs) [41]: zk-SNARKs allow for the verification of computations without revealing the inputs or intermediate steps. This technology can be used in blockchain to prove the validity of a computation, such as verifying a smart contract, while keeping the inputs confidential.

7.  Permissioned/Private Blockchains [42]: Permissioned or private blockchains restrict participation and access to a select group of known entities. These blockchains provide enhanced privacy as they limit the visibility of transactions and data to authorized participants.

Figure 2 presents an extensive blockchain architecture standardization that can be applied to several novel industrial applications [4].

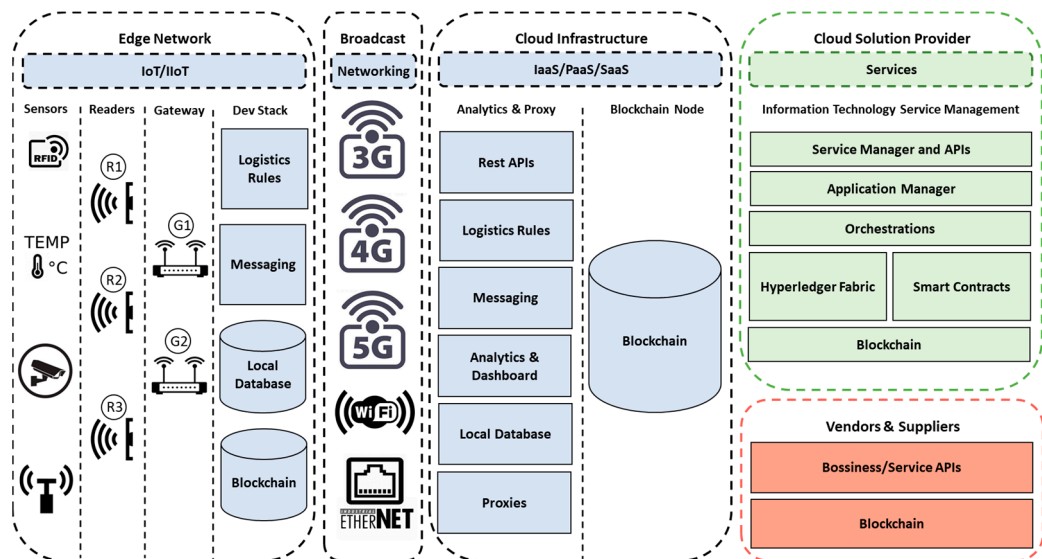

**Figure 2.** Blockchain scalable architecture for industrial ecosystems [4].

Here are a few examples of how blockchain technology has been applied in various scenarios to enhance privacy:

1.  Healthcare Data Sharing: Blockchain can be used to improve the privacy and security of healthcare data sharing. By storing medical records and sensitive patient information on a blockchain, access can be controlled, and data integrity can be ensured. Patients have control over their own data and can grant access to healthcare providers as needed, reducing the risk of unauthorized access or data breaches. One example is MedRec, a blockchain-based system that allows patients to securely share their medical records with healthcare providers while maintaining privacy and data ownership.

2.  Supply Chain Management: Blockchain technology has found applications in enhancing privacy and transparency in supply chain management. By recording transactions and tracking products on a blockchain, stakeholders can verify the authenticity and provenance of goods without revealing sensitive business information. This helps prevent counterfeit products and provides transparency for consumers. IBM's Food Trust is a notable example that utilizes blockchain to track and trace food products, ensuring the integrity of the supply chain and providing consumers with information about the origin and handling of their food.

3.  Identity Management: Blockchain offers potential solutions for secure and privacy-preserving identity management systems. By using blockchain, individuals can maintain control over their personal data and selectively disclose information to third parties, reducing the risk of identity theft and unauthorized data access. Self-

sovereign identity (SSI) solutions, such as uPort and Sovrin, leverage blockchain to enable individuals to manage and control their digital identities, providing privacy-enhancing features and reducing reliance on centralized identity providers.

4.  Financial Transactions and Privacy: Blockchain technology can improve privacy in financial transactions by reducing the need for trusted intermediaries and providing pseudonymity. Cryptocurrencies like Bitcoin and privacy-focused cryptocurrencies like Monero utilize blockchain to facilitate secure, decentralized, and pseudonymous transactions. While blockchain transactions are public, privacy-focused techniques such as ring signatures, stealth addresses, and zero-knowledge proofs are employed to obfuscate transaction details and enhance privacy.

It is worth noting that while blockchain technology can enhance privacy in these scenarios, its implementation requires careful consideration of the specific use case, including factors such as regulatory compliance, scalability, and user adoption.

Federated learning is also a privacy-preserving architecture that enables collaborative machine learning on decentralized data. It allows multiple parties, such as individual devices or edge servers, to train a shared machine learning model without directly sharing their raw data with a central server or each other. Here are some key aspects of federated learning that contribute to its privacy-preserving nature [43,44]:

1.  Local Training: In federated learning, the training of the machine learning model takes place locally on individual devices or edge servers. This means that data remains on the devices, and only model updates (such as gradients) are shared with the central server or aggregator.

2.  Differential Privacy: Differential privacy techniques can be employed in federated learning to further protect privacy. By adding controlled noise or perturbation to the local model updates before sharing them, the individual data points and patterns are obfuscated, preventing the reconstruction of sensitive information.

3.  Encryption: Encryption techniques can be applied to protect the confidentiality of the model updates during transmission. Secure multi-party computation (MPC) protocols, homomorphic encryption, or secure enclaves (such as Trusted Execution Environments) can be utilized to ensure that the model updates remain private.

4.  Aggregation with Privacy Preservation: The central server or aggregator collects the encrypted or differentially private model updates from the participants and performs the aggregation to update the shared model. Aggregation techniques can be designed in a way that preserves privacy, such as using secure aggregation protocols that do not reveal individual contributions.

5.  On-Device Personalization: Federated learning can also support on-device personalization, where the shared model is further fine-tuned or customized on individual devices using locally available data. This approach ensures that sensitive data remains on the user's device, enhancing privacy.

6.  Secure Communication: Secure communication protocols, such as encrypted channels and secure socket layers (SSL/TLS), should be employed during data transmission between the participants and the central server to protect against eavesdropping and data tampering.

Federated learning allows organizations or individuals to leverage the collective intelligence of decentralized data while minimizing the risks associated with data sharing. This architecture promotes privacy by keeping sensitive data localized, incorporating privacy-preserving algorithms, and utilizing encryption and secure communication protocols. Figure 3 presents the Federated Auto-Meta-Ensemble Learning (FAMEL) architecture in the new IT/OT industrial environment [45].

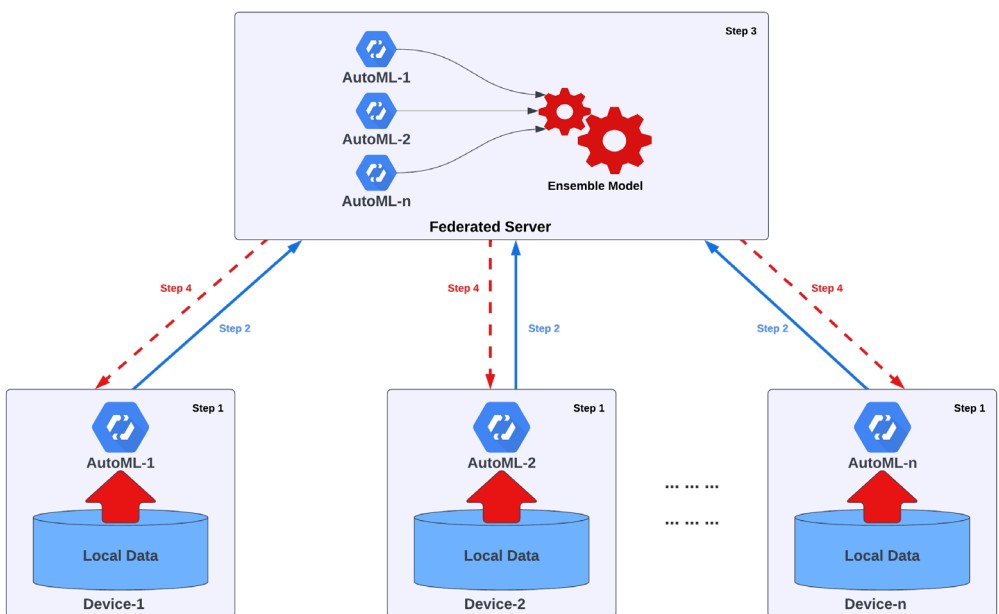

**Figure 3.** Basic security reference architecture in the industrial environment [45].

It is a holistic system that automates selects and uses the most appropriate algorithmic hyperparameters to optimally solve a problem under consideration, approaching it as a model for finding algorithmic solutions where it is solved via mapping between the input and output data. The proposed framework uses meta-learning to identify similar knowledge accumulated in the past to speed up the process. This knowledge is combined using heuristic techniques, implementing a single, constantly updated intelligent framework. The data remains in the local environment of the operators, and only the parameters of the models are exchanged through secure processes, thus making it harder for potential adversaries to intervene with the system.

Here are a few examples of how federated learning technology has been applied in different scenarios to enhance privacy:

1.  Healthcare [46]: Federated learning can be applied in healthcare settings to enable collaborative model training while preserving patient privacy. Hospitals or medical institutions can train machine learning models using local patient data without sharing sensitive patient information. The models are then aggregated or updated in a privacy-preserving manner, allowing healthcare providers to benefit from shared insights without compromising patient confidentiality. This approach can be useful for applications such as disease diagnosis, medical image analysis, or predictive analytics.

2.  Smart Devices and the Internet of Things (IoT) [47]: Federated learning is well-suited for scenarios involving edge devices or IoT devices. These devices often have limited computational resources, making it challenging to send large amounts of data to a centralized server for model training. With federated learning, edge devices can collaboratively train machine learning models using locally collected data while keeping the data on the device. Only model updates or aggregated information is sent to a central server, ensuring privacy while benefiting from shared knowledge. This is useful in applications such as personalized recommendations, activity recognition, or anomaly detection in smart homes or industrial IoT settings.

3.  Financial Services [48]: Federated learning can enhance privacy in financial services by enabling collaborative model training while keeping sensitive customer data on local servers or devices. Banks or financial institutions can train machine learning models for tasks like fraud detection or credit scoring using locally held customer data. The models' updates or aggregated information are exchanged in a privacy-

preserving manner, ensuring the privacy of individual customer transactions and sensitive financial information.

4. Natural Language Processing (NLP) [49]: Federated learning can be applied in NLP tasks to protect user privacy while improving language models. Instead of centralizing user data on a single server, federated learning allows individual devices or servers to train language models using local data. The models' updates or aggregated information, which preserve the privacy of individual texts, are shared across devices or servers. This approach enhances privacy while enabling the improvement of language models for applications such as voice assistants, chatbots, or sentiment analysis.

These examples illustrate how federated learning can be leveraged in various domains to enable collaborative model training while maintaining privacy and data confidentiality. By keeping the data decentralized and only exchanging model updates or aggregated information, federated learning offers a privacy-enhancing approach for machine learning in scenarios where data privacy is crucial.

A discussion and comparison of these two approaches (Blockchain and Federated Learning) are presented in the Table 2.

**Table 2.** Blockchain vs. Federated Learning.

| Technology | Description | Privacy Benefits | Challenges | Comparison |
|---|---|---|---|---|
| Blockchain | Blockchain technology is a decentralized and distributed ledger system that offers enhanced security and privacy features. It ensures the integrity and immutability of data by storing transactions in a chain of blocks, making it difficult for malicious actors to alter or tamper with the data. | Data Transparency: Blockchain allows participants in the network to have access to a transparent and auditable history of transactions without revealing specific identifying information. | Scalability: Blockchain networks can face challenges in terms of scalability due to the consensus mechanisms and the need to replicate data across multiple nodes, resulting in slower transaction speeds. | Data Handling: Blockchain technology stores data directly on the ledger. |
| | | Data Integrity: The decentralized nature of blockchain ensures that data stored on the ledger is tamper-resistant, making it difficult for unauthorized parties to modify or manipulate information. | Energy Consumption: Some blockchain networks, particularly those utilizing proof-of-work consensus, require significant computational power, leading to high-energy consumption. | Data Privacy: Blockchain provides transparency and integrity but may not provide strong privacy for data contents. |
| | | Secure Transactions: Blockchain employs cryptographic techniques, such as digital signatures and encryption, to secure data transfers and ensure authenticity. | Data Privacy: While blockchain technology ensures data integrity and immutability, it does not inherently provide strong privacy protection for the contents of the data. The transparency of blockchain can potentially reveal sensitive information about transactions. | Trust Model: Blockchain is based on a decentralized trust model. |

**Table 2.** *Cont.*

| Technology | Description | Privacy Benefits | Challenges | Comparison |
|---|---|---|---|---|
| Federated Learning | Federated learning is an approach where machine learning models are trained across multiple decentralized edge devices or servers without sharing the raw data. Instead, only model updates or aggregated information is exchanged between the devices and a central server, ensuring data privacy. | Data Localization: Federated learning allows data to remain on local devices or servers, reducing the risk of data breaches or unauthorized access. | Model Heterogeneity: Federated learning can be challenging when dealing with a diverse range of edge devices or servers with different computational capabilities, data distributions, or data quality. | Data Handling: Federated learning keeps the data locally and only exchanges model updates or aggregated information. |
| | | Privacy-Preserving Model Training: The model updates or aggregated information shared during federated learning are typically anonymized and encrypted, preserving the privacy of individual data points. | Central Server Trust: While federated learning aims to preserve privacy, it still requires trust in the central server that aggregates model updates. A compromised or malicious server could potentially extract sensitive information from the updates. | Data Privacy: Federated learning focuses on preserving the privacy of individual data points during model training. |
| | | Reduced Data Transmission: Federated learning minimizes the need to transfer large amounts of raw data to a central server, which can be beneficial in bandwidth-constrained environments or when dealing with sensitive data. | Model Interpretability and Debugging: Federated learning can make it challenging to interpret and debug models trained across multiple devices or edge nodes. Understanding the reasons behind model performance issues, identifying erroneous contributions, or diagnosing the root causes of failures may require specialized techniques and tools. | Trust Model: Federated learning relies on trust in the central server and the integrity of participants. |

It is important to note that these solutions are not mutually exclusive, and their applicability depends on the specific requirements and constraints of the IIoT ecosystem. Organizations may choose to combine these approaches or utilize other privacy-enhancing technologies to achieve the desired level of privacy and security. This paper aims to provide insights into the privacy dimensions of the IIoT and provide recommendations for ensuring secure and privacy-respecting implementation, which can help to build trust in IIoT systems and ensure their successful adoption.

**4. Dimensions of Privacy**

Privacy in the industrial sector is a concept that is very difficult to define, especially in the digital age of the IIoT, where the convergence of services creates unclear boundaries of definition. Depending on the contexts, relationships, and products involved, it can have many connotations. To properly design privacy settings for IIoT architectures, technologists

must research and understand the dimensions of privacy that are important to the users of the services in question [27].

Privacy, as derived from E.U. directives and based on the way it is described by Martınez-Ballester et al. [50], can be categorized as or take the following dimensions:

1.  Identity Privacy. It concerns the identity details of an entity and is related to the concepts of authentication and authorization. Most IIoT data is designed for usage by restricted user groups [51]. Consequently, authentication (understanding the identity of the node or user) and authorization (by granting the necessary access rights) are critical, especially regarding copyright, patents, etc., which are crucial to the existence and viability of the industry [52].
2.  Location Privacy. It refers to an entity's location identification information. Said determination violates personal or industrial privacy issues concerning the detection, identification, storage, processing, and sharing of information in a technical or legal context [53].
3.  Footprint Privacy. It refers to an entity's unique traceable communication actions. A feature of this function can be found in smart energy grids, characterized by real-time two-way communications [54]. The difficulty of controlling and retrieving energy data exchanged with third parties threatens network users' privacy. The proposed approach is a comprehensive solution for overcoming concerns related to Wen et al. The standard of footprint's privacy allows a home user to save encrypted measurement data on a cloud server. When financial audits are required, an authorized requester can submit two queries to the cloud server to retrieve the measurement data [55].
4.  Multidimensional Privacy. It refers to an entity's multidimensional or complex identification element, which may combine some of the above dimensions [50]. Solving such problems requires complex integrated processes or solving and parameterizing the individual issues in a custom schema [56].

These dimensions provide a framework for understanding the privacy needs of IIoT ecosystems and guiding the development of privacy settings and strategies. It is important to note that while these dimensions are specific to the industrial domain, they align with the broader principles of data protection and privacy regulations, such as the General Data Protection Regulation (GDPR). It should be noted that no relevant regulation exists for protecting persons' privacy, exclusively for the industrial domain. For the processing of personal data in the industrial domain and the free movement of such data by competent industrial authorities for the prevention, felicitation, detection, or prosecution of criminal offenses or the execution of sanctions, industries follow the existing regulatory offenses [57], where almost every industry is involved in processing personal data in one or more processes [58].

The industrial domain often requires specific privacy regulations due to its unique characteristics and privacy concerns that may not be fully addressed by the existing general regulations, like GDPR [59–61]. Here are some reasons why the industrial domain may benefit from specific privacy regulations:

1.  Specialized Data Types: The industrial domain deals with specific types of data that may not be explicitly covered by the general regulations. Industrial environments often involve sensitive data related to manufacturing processes, proprietary technologies, trade secrets, industrial control systems, or safety protocols. These data types require specialized privacy considerations to protect intellectual property, ensure operational safety, and prevent unauthorized access or misuse.
2.  Complex Data Ecosystems: Industrial environments typically have complex data ecosystems with interconnected machines, sensors, and control systems. These systems generate and exchange vast amounts of data, often in real-time. General regulations like GDPR may not adequately address the intricacies of managing and securing data within such heterogeneous and dynamic environments. Specific regulations can provide guidelines and requirements tailored to the unique challenges of industrial data ecosystems.

3. Safety and Security Risks: Privacy regulations in the industrial domain need to consider not only the protection of personal data, but also the safety and security risks associated with industrial processes. Data breaches or unauthorized access in industrial settings can have severe consequences, including physical harm, environmental damage, or disruptions to critical infrastructure. Specific regulations can address these risks and impose safeguards to mitigate these potential threats.

4. Industry-Specific Compliance Requirements: Different industries within the industrial domain, such as manufacturing, energy, or healthcare, may have specific compliance requirements related to privacy and data protection. These requirements may be driven by sector-specific regulations, standards, or contractual obligations. Specific privacy regulations can align with these industry-specific compliance requirements to ensure that organizations within the industrial domain adhere to the necessary privacy practices.

5. Operational Constraints and Challenges: The industrial domain often faces operational constraints and challenges that are distinct from other sectors. These may include limited connectivity, remote locations, harsh environments, or legacy systems. General privacy regulations may not consider these operational constraints, making it necessary to have specific regulations that accommodate the unique circumstances of the industrial domain while still ensuring data privacy.

It is important to note that while general regulations like GDPR provide a valuable framework for data protection and privacy, the industrial domain may require additional measures and specific regulations to address its unique characteristics, data types, ecosystem complexity, safety risks, compliance requirements, and operational challenges. The specific regulations can build upon and complement the general regulations, providing a more comprehensive and tailored approach to privacy in the industrial context.

As follows from the above-mentioned text, privacy dimensions refer to the different aspects of privacy that need to be considered in IIoT implementations. In the context of privacy dimensions, the privacy needs of an IIoT ecosystem are investigated as specified by industry standards. In conclusion, the study identifies four dimensions of privacy that are relevant to the IIoT:

1. Data Protection: This dimension refers to the protection of personal data and the need to ensure that data is collected, stored, and processed in a way that complies with applicable data protection laws and regulations.

2. Confidentiality: This dimension protects sensitive information from unauthorized access or disclosure. In IIoT systems, confidentiality is particularly important for ensuring the security of trade secrets, intellectual property, and other proprietary information.

3. Availability: This dimension ensures that IIoT systems and devices are available and operational when needed. Availability is essential to ensure critical infrastructure and services are not disrupted or compromised.

4. Integrity: This dimension refers to the need to ensure the accuracy and reliability of IIoT data and systems. In IIoT systems, integrity is particularly important for ensuring that decisions based on data are accurate and malicious actors do not compromise the systems.

5. Non-repudiation: It is a security property that ensures the sender of a message or a digital transaction cannot later deny sending the message or engaging in the transaction. It provides proof that a particular action or communication occurred and prevents individuals from disowning their actions or denying their involvement. In the context of privacy, non-repudiation plays a significant role in maintaining trust and accountability in digital interactions. Specifically:

   a. Digital Transactions: It ensures that both parties involved in a transaction cannot later deny their participation, protecting the privacy of sensitive information exchanged during the transaction.

　　b.　　Message Authenticity: Non-repudiation guarantees that the sender of a message cannot deny sending it. This property is particularly essential when exchanging private or confidential messages. It helps prevent unauthorized access and ensures that the recipient can trust the origin and authenticity of the message.

　　c.　　Digital Signatures: Digital signatures are a cryptographic mechanism used for non-repudiation. By using digital signatures, individuals can sign electronic documents or messages, providing assurance that the content remains unchanged and that the signer cannot later deny their approval.

　　d.　　Legal Implications: Non-repudiation can have legal implications in contracts and agreements. If a digital transaction or communication has non-repudiation measures in place, it can serve as evidence in case of disputes, protecting the privacy of individuals involved by establishing their roles in the interaction.

By identifying these dimensions of privacy, the study aims to provide a framework for understanding the privacy needs of IIoT ecosystems and developing effective strategies for ensuring privacy and security in industry implementations.

## 5. Industrial Privacy

According to Boussada et al. [62], privacy in the context of the IIoT refers to an individual's right to control and influence the collection, retention, and disclosure of their personal information. Privacy becomes a significant concern in the IIoT environments due to the extensive collection, processing, and sharing of data, which raises serious issues related to its privacy implications [63].

In addition to personal privacy, data protection is a crucial challenge faced by businesses in various industries, particularly when handling sensitive information such as financial data. Industrial privacy refers to the protection of sensitive information and trade secrets within the context of industrial and manufacturing sectors. It involves safeguarding confidential data, proprietary processes, intellectual property, and other critical business information from unauthorized access, theft, or misuse by competitors, employees, or external entities.

Regulatory authorities emphasize good data management practices to enhance customer profiling, identify potential opportunities, and conduct risk management analysis. Privacy and data protection need to be monitored throughout the client lifecycle, and this extends to multiple use cases in the industrial sector, such as data exchange for customer protection, improved credit risk assessment, and secure claims management [64,65].

While leaders in industries like manufacturing or heavy industry may not prioritize data privacy as highly as other sectors, the digital age is changing this perspective. Data privacy has become a significant risk for any industry that deals with potentially sensitive data concerning customers, employees, and business partners [66].

The complexity of the IIoT ecosystem and the need for regulatory compliance at both the hardware and software levels pose significant challenges in protecting these systems. In the IIoT ecosystem, where multiple systems interact with the physical world, the uncontrolled arrangement of states can lead to dangerous conditions. This necessitates multi-layered security approaches and encryption techniques to address privacy concerns. The convergence of computer and communication technologies, decentralization of processing, and distributed analysis in industrial activities further complicates the concept of privacy and gives rise to new challenges [67,68].

In this context, the need to protect industrial privacy becomes crucial. Existing regulations and standardization efforts are often insufficient to provide a robust shield against emerging threats. Innovative solutions, such as a central Software-Defined Network (SDN) control layer, deterministic scheduling, and lightweight encryption, can offer improved privacy-preserving IIoT standardization systems [69].

Furthermore, it is essential to continuously update and adapt the existing standards and systems used in industrial technologies to enhance privacy and security [70,71]. The surveillance of key infrastructures and the control of certification and identity play signifi-

cant roles in maintaining privacy in the IIoT environment [72,73]. Sharing best practices, recommendations, and knowledge for security maintenance and continuous improvement is also vital in ensuring privacy in the evolving IIoT landscape [74].

Industrial privacy focuses on safeguarding confidential, proprietary, and sensitive information related to industrial operations, such as trade secrets and intellectual property. Compliance with data protection laws, such as the GDPR, is crucial in collecting and processing personal data in industrial settings [75]. It involves implementing technical and organizational measures to protect sensitive information, training employees on their responsibilities, and regularly reviewing and updating privacy policies and procedures to address emerging threats and risks.

In summary, industrial privacy plays a vital role in preserving the confidentiality, integrity, and availability of sensitive information in industrial settings. It requires a comprehensive approach that addresses the various aspects of privacy, including data protection, confidentiality, availability, and integrity. By implementing appropriate measures, regularly updating policies, and ensuring employee awareness, organizations can ensure responsible data use while safeguarding the rights and privacy of the individuals and entities involved in industrial operations.

## 6. Privacy Threats in the IIoT

IIoT solutions provide numerous advantages to organizations, including enhanced operational efficiency, reduced costs, and increased productivity. Through task automation, streamlined processes, and effective resource management, organizations can optimize workflows, minimize downtime, and achieve higher levels of overall efficiency. Leveraging the data collected from connected devices and sensors, organizations can make informed decisions, identify and address bottlenecks, and optimize resource allocation. The results are improved productivity, cost reduction, and a competitive edge in the market [76]. However, IIoT devices are as susceptible as any other Internet-connected device [77]. Adopting more powerful and sophisticated equipment, such as microprocessors, makes it difficult to address cybersecurity and privacy concerns [78]. Examples of attacks on Industrial Control Systems (ICS) include distributed control systems, programmable logic controllers, supervisory control and data acquisition, and Human-Machine Interfaces (HMI) [79].

ICS and HMI can be vulnerable to various types of attacks, which can have severe consequences on industrial operations. For example [26,80]:

1. Malware and Ransomware: Attackers can deploy malware or ransomware specifically designed to target ICS or HMI systems. These malicious programs can disrupt the functioning of critical industrial processes, compromise system integrity, and even demand ransom payments in exchange for restoring normal operations.
2. Distributed Denial of Service (DDoS): A DDoS attack involves overwhelming a system with a flood of traffic, causing it to become unresponsive or crash. If an attacker successfully launches a DDoS attack against an ICS or HMI system, it can disrupt control signals, delay response times, or render the system inoperable, leading to production interruptions or safety risks.
3. Unauthorized Access and Control: If an attacker gains unauthorized access to an ICS or HMI system, they can manipulate control parameters, change setpoints, or issue unauthorized commands. This can lead to process deviations, equipment damage, safety hazards, or even catastrophic incidents.
4. Insider Threats: Insiders with malicious intent, such as disgruntled employees or contractors, can abuse their privileged access to ICS or HMI systems. They may deliberately tamper with control settings, sabotage equipment, or steal sensitive data, causing significant disruptions or compromising system integrity.
5. Social Engineering: Attackers may employ social engineering techniques to trick authorized users into divulging sensitive information or granting unauthorized access. For example, phishing emails or phone calls can deceive employees into revealing

login credentials or executing malicious commands, which can be used to compromise ICS or HMI systems.

6.  Supply Chain Attacks: ICS or HMI systems can be targeted through vulnerabilities in the supply chain. Attackers may compromise the integrity of hardware, software, or firmware during the manufacturing, distribution, or installation process. This can result in the introduction of malicious components or exploitable weaknesses in the system.

The consequences of these attacks can range from operational disruptions and financial losses to safety incidents and environmental hazards. Protecting ICS and HMI systems requires robust security measures, including network segmentation, regular software updates and patches, strong access controls, user training, intrusion detection systems, and incident response plans.

It is important to note that the examples provided are not exhaustive, and the specific attack vectors and techniques can vary depending on the specific ICS or HMI system in use. Organizations need to assess and mitigate risks based on their unique operational requirements and the potential threats they face.

Grouping vulnerabilities by categories aids in establishing and putting into practice mitigation measures because all ICS deal with information technology (IT) and operational technology (OT). These categories are divided into policy and practice concerns and vulnerabilities found in various platforms (such as hardware, operating systems, and ICS applications). Vulnerabilities can be grouped as Policy Concerns, Practice Concerns, Hardware Vulnerabilities, Operating System Vulnerabilities, ICS Application Vulnerabilities, etc. These categories can help in establishing mitigation measures and prioritizing security efforts [81].

Every ICS environment could have flaws depending on how they are set up and what they are intended to do. A further consideration is an ICS environment's size; the larger the environment, the higher the likelihood that an error could occur. An ICS environment that has modernized its outdated systems and added new technologies like Industrial Internet of Things (IIoT) devices might also have more vulnerabilities that threat actors might take advantage of.

Figure 4 depicts a blockchain-based system that combines a multi-signature system and differential privacy algorithms to provide a secure and privacy-preserving solution for a network traffic analysis called BANTA [82,83].

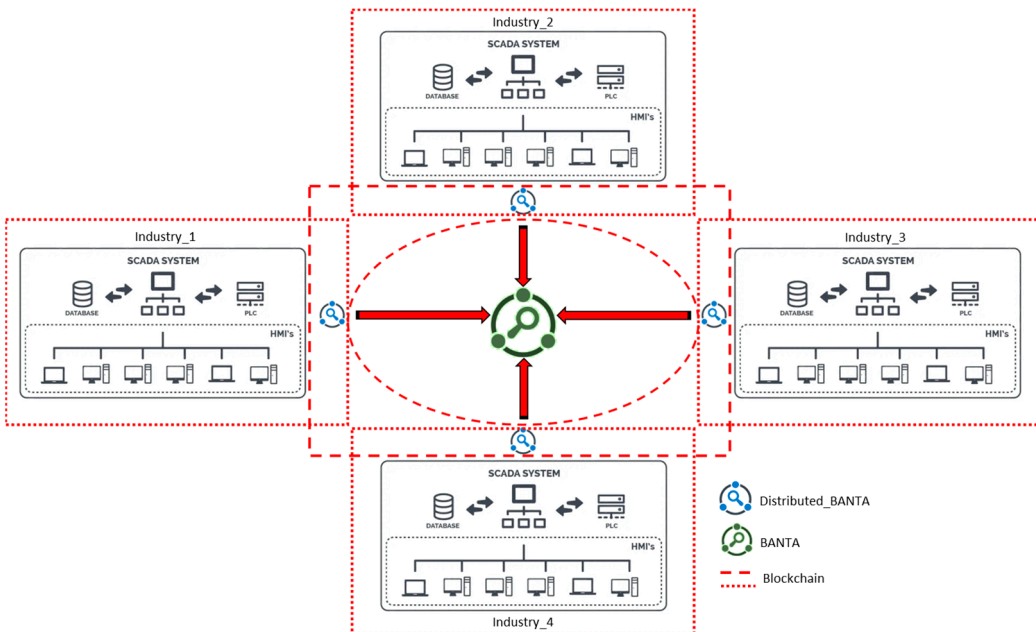

**Figure 4.** Use case of BANTA architecture in the ICS network [82].

The BANTA system employs smart contracts to automate network traffic analysis and store network traffic logs in a decentralized manner, resulting in increased transparency and security in the ICS. This multi-signature system requires a specific set of parties to sign off on any changes to the smart contract's behavior or data, preventing rogue administrators from tampering with the smart contract and jeopardizing its security or reliability. In addition, it uses differential privacy algorithms to protect sensitive information in network traffic logs while allowing network administrators to analyze network traffic.

The main concerns regarding protecting data privacy in industrial domains are intertwined with the more general concerns about the risks associated with each modern network device [84]. In general, the most basic and common privacy threats related to the IIoT are:

1. Identification and Authorization. It is directly related to the concept of identity privacy. It refers to the effort to find correlations between the data that can be used to detect, identify, and maliciously replicate the application of profiles (sets of associated data) to personalize and remember secret, industrial information. Techniques such as the Subscriber Identity Module (SIM) and the Machine Identification Module (MIM), proposed by Borgia [51], are essential solutions worthy of attention. However, these approaches work in centralized single-management networks. At the same time, it is not easy in distributed topologies to manage identification services and standardizations such as the one proposed by Moosavi et al. [52], and it concerns an architecture of authorization of remote end-users using distributed smart gateways, which are based on the Datagram Transport Layer Security (DTLS) handshake protocol. In addition, attacks on the IIoT compromise authorized industrial systems access, and as a result, one such security issue can degrade the related services. Ransomware also causes IIoT devices to malfunction and steals users' sensitive information and data. In addition, if many smart IIoT devices cannot encrypt user data, malware will emerge. IIoT devices use a network that does not convert data into code to prevent unauthorized device access.

2. Localization and tracking. It is directly related to the concept of location privacy [85]. An industry can choose the locations that it chooses to perform its economic functions in. Several issues influence the choice of a suitable location, most notably the nature and characteristics of the industrial activity carried out by the enterprise (e.g., extraction of raw materials or cultivation, production of intermediate or final products, provision of a service) and the associated costs of production, balanced with the cost of physical distribution to the target markets and the importance of proximity to customers as a basis for establishing competitive advantages over rival suppliers. Some locations may be preferred for their production advantages, for example, due to lower labor costs, the availability of investment subsidies, the supply of skilled workers, and parallel access to relevant facilities. Similarly, many service activities must be located in and around the customer's catchment areas. At the same time, some suppliers may be interested in operating alongside their core customers to synchronize production input requirements better.

3. On the other hand, the high price of distribution, especially in the case of bulky products with low added value or the international context, the imposition of tariffs and quotas on imports creates essential requirements for an appropriate position oriented to the market, but one that is also protected from the prying eyes of the competition and espionage. A low-cost technical solution that adds protection to the IoT environment was proposed by Joy et al. [53] by embedding in GPS devices privacy software that ensures that IoT devices and their administrators have fine-grained control over releasing their position. In addition, the safety of the data ingested from numerous IIoT devices is related directly to other data security and privacy concerns from insecure cloud infrastructures, web applications, and mobile environments. As a result, it is necessary to follow data transmission security rules in each domain so that there are measures in place to identify the path from whose device the data is

transmitted. It is also critical to eliminate irrelevant data and data without relation to the actual operation. Although compliance with numerous regulatory structures becomes complex when multiple data is stockpiled, the infrastructure must be carried out with separate services for controlling the data linked to interconnected devices and environments [86].

4.  Profiling. The threat lies in violating privacy and monitoring persons or individuals in their association with specific industrial processes. Accordingly, it may refer to identifying, collecting, and processing information derived from services or reference models, which may constitute an industrial secret. Characteristics of the ongoing concern for protecting IoT devices from profiling threats are efforts to enhance privacy in RFID devices [87,88], sensor systems [89], wireless networking [90,91], and identity management [88,92] technologies, to enhance privacy or encryption technology [93,94].

5.  Hardware Lifecycle. Industrial devices are, in most cases, remanufactured and reused. Therefore, sensitive information, device logs, and data stored in memories or storage media will likely fall into the wrong hands with unpredictable consequences [95]. For the specific threats, the industry should draw up and implement a uniform policy for the management of industrial equipment, as well as apply techniques of total deletion [96] of the data locally or in distributed information processing systems which include first and second sites, which may consist of corresponding information production and copying sites [97]. Also, IIoT hardware addresses security and privacy threats from inadequate testing and a lack of upgrading processes [98]. IIoT device manufacturers, while willing to produce various devices, do not consider the security and upgrading concerns of said devices because they require extensive testing and, therefore, additional costs. These malfunctions increase the possibility of security and privacy attacks when released into a real-world industrial infrastructure [96].

6.  Inventory attack. Inventory assaults are the unlawful acquisition of information about the equipment's presence and attributes. Also, with the implementation of machine to machine vision [99,100], intelligent devices can be questioned about their energy footprint, communication rates, reaction times, and other distinctive characteristics, which might be used to identify their kind and model, subject to limitations imposed by legal or presumably legal organizations. Thus, evil individuals who violate the privacy of an industry can assemble an inventory list of the gadgets in a particular building or factory, as well as information about how each device operates [17]. Here too, cryptography solutions have been proposed for aggregation mechanisms. This secure aggregation protocol meets the IoT requirements [101]. It evaluates its effectiveness in light of various system configurations, the wireless channel's impact on packet error rates, and private communication techniques [102].

7.  Linkage. This threat consists of connecting different previously separated systems so that combining the data and the sources reveals critical information that would be impossible to tell by individual plans. Moreover, to ensure the smooth operation of IIoT devices, it is essential to have flat networking that will allow them to function effectively. It is critical to have a high-quality open networking system for this purpose [103]. This particular factor in IIoT networks creates a security barrier. In this regard, industrial enterprises must thoroughly assess their security policies to ensure that IIoT devices are not vulnerable to threats [68]. Also, providers must understand the significance of adequately configuring the networking device and services and that data privacy entails various processes, such as efficiently removing sensitive information through data segregation [48].

Figure 5 depicts an emerging architecture that successfully predicts and assesses threat-related conditions in an industrial ecosystem while ensuring privacy and secrecy [104].

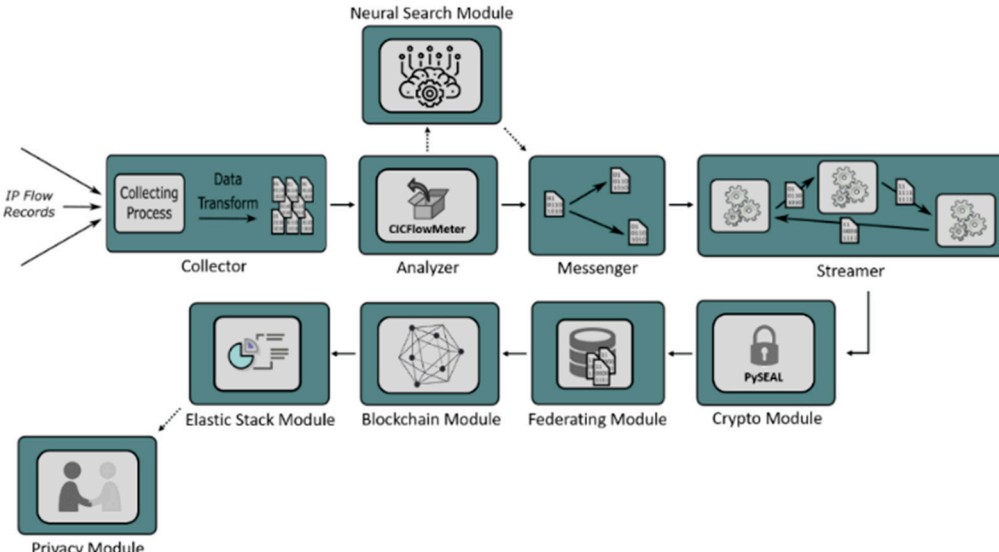

**Figure 5.** Privacy-preserving industrial framework [104].

The above high-level depiction of the specific industrial privacy-preserving framework introduces an intelligent control mechanism to detect abnormalities in the Industry 4.0 communication network based on three main principles. The sensitive data is not transmitted through communication channels or stored in a central point of attack, and the learning algorithms are constantly updating their predictive power. However, a significant part of the responsibility lies with the hardware manufacturers. Often, the mechanisms and interoperability standards concerning the security of IIoT devices should be addressed or treated as a secondary consideration [105]. Usually, this is due to the requirement for a short period to implement an IoT device, simplify the design of its operating mechanisms, and reduce its overall cost. Therefore, those involved in IIoT delivery processes must consider privacy and develop privacy management interfaces built into the endpoint and web interface of the product or service [106]. This technology should enable the user industry to understand which privacy features are utilized by the ecosystem, what the Terms of Service are, and whether or not it is feasible to deactivate the disclosure of this information to the business' partners or rivals. This information management system will ensure that users have the right and capacity to regulate the information they disclose about themselves and their physical surroundings [107].

In light of the above-mentioned text, the IIoT presents unique privacy challenges and threats due to the large amounts of data generated, stored, and processed in industrial environments. Some of the most common privacy threats in the IIoT include Data Breaches, Cyberattacks, Insider Threats, Lack of Transparency and Unsecured Communications.

To address these privacy threats, IIoT implementations must implement appropriate technical and organizational measures to protect sensitive information and personal data. These measures may include encryption, access controls, employee training, regular vulnerability assessments, and ongoing monitoring of IIoT systems and devices. By implementing these measures, IIoT systems can better protect against privacy threats and ensure confidentiality, integrity, and availability of sensitive information.

## 7. Privacy Requirements

Privacy requirements involve complying with relevant privacy laws, industry policies, notices, and contractual obligations related to the gathering, recording, usage, storage, processing, sharing, safeguarding, security (technical, physical, and administrative), disposal, destruction, disclosure, or transfer (including cross-border) of sensitive personal data [108]. But because something so strict in an industrial environment is practically impossible to

implement, there are basic requirements and recommendations that should be followed in an industrial setting to respects privacy, as listed in the following:

1.  Mitigation by Design. During the design process of IoT goods, services, or systems, privacy protection measures should ideally be planned and implemented [109,110].
2.  Assessment. Privacy impact assessments should help provide a secure method of analyzing how personally identifiable information is collected, stored, secured, shared, and handled and how it is disposed of, governed by this section [105,111,112].
3.  Legal Compliance. To monitor compliance, applicable legal or regulatory requirements should be assessed [113–115].
4.  Use Limitation. A provision should be made for the necessary work to ensure that access to any physical or electronic security system is restricted to fully authorized persons and for fully authorized purposes [116–118].
5.  Storage Safeguards. Warehouses, data lakes, and databases, where personal information is collected and stored, should be protected in terms of physical and logical security [119–123].
6.  Secure Communications. The data transmitted between systems or components, and more generally, communications in an IoT environment, should be protected from unauthorized disclosure or access [124–127].
7.  Transparency. Individuals whose personal information may be collected should be notified of the reason for collection and how that information may be used. There should also be mechanisms that can reveal possible personal data leaks [128–130].
8.  Data Retention Policy. There should be a policy that defines the retention period of personal data, the methods of destruction of such data, and a procedure that ensures that deleted information is not recoverable [131–133].

## 8. Suggestions

To address privacy threats in the IIoT, it is important to establish clear privacy requirements and implement appropriate measures to ensure compliance with applicable data protection laws and regulations. Some privacy requirements and suggestions for IoT implementations include:

1.  Data Protection Impact Assessments (DPIAs): Conducting DPIAs can help identify potential privacy risks and ensure that appropriate measures are in place to mitigate these risks.
2.  Privacy by Design: Implementing privacy by design principles can help ensure that privacy is integrated into developing IIoT systems and devices from the outset.
3.  Encryption and Access Controls: Encryption and access controls can help protect sensitive information and personal data from unauthorized access or disclosure.
4.  Regular Vulnerability Assessments: Regular assessments can help identify and address security and privacy vulnerabilities in IoT systems and devices.
5.  Employee Training: Ensuring that employees are trained in privacy requirements and best practices can help minimize the risk of insider threats and improve the overall privacy posture of the organization.
6.  Compliance with Applicable Data Protection Laws and Regulations: Compliance with applicable data protection laws and regulations, such as the GDPR [134] in the European Union, is essential for protecting the privacy rights of individuals and organizations involved in IIoT operations.
7.  Regular Review of Privacy Policies and Procedures: Regularly reviewing and updating privacy policies and procedures can help ensure that they are up-to-date and effective in addressing emerging threats and risks.

By implementing these privacy requirements and suggestions, IIoT implementations can better protect against privacy threats and ensure the safe and responsible use of technology and data in industrial settings. In addition, the industrial partners will enable disruptive business models for personalization and full automation of secure and private processes.

## 9. Discussion

The adoption of the secure and privacy-enhanced Industrial Internet of Things (IIoT) in Industry 4.0 solutions presents significant challenges for businesses, as they strive to seamlessly integrate emerging technologies into their production processes. This challenge primarily revolves around the readiness and maturity of companies to incorporate robust security technologies within their production infrastructure, encompassing both secure infrastructure and privacy-preserving services. By successfully adopting privacy-preserving IIoT technologies, businesses can establish the foundation for flexible manufacturing, enabling them to promptly respond to dynamic market changes and demands.

The research findings offer a comprehensive overview of the IIoT and its integration with intelligent systems within the Industry 4.0 ecosystem. It sheds light on the prevalent security issues faced by IIoT deployments, emphasizing the crucial need for privacy protection within these systems. The paper meticulously examines how industry norms can be met in terms of privacy requirements for IIoT environments. Furthermore, it provides an in-depth analysis of current strategies and solutions for effectively managing privacy risks within industrial settings [135].

The paper's focal point on privacy concerns within the IIoT ecosystem, along with its valuable recommendations for addressing privacy risks, significantly contributes to the existing literature on this subject. By comprehensively addressing privacy concerns and security risks, the paper adopts a more holistic approach towards ensuring the safety and privacy of the IIoT ecosystem and its end-users.

For organizations, the paper provides valuable insights into the privacy risks associated with the IIoT and recommendations for addressing those risks. By implementing the strategies outlined in the paper, organizations can better protect sensitive information and ensure compliance with data protection regulations and other applicable laws. This can help organizations maintain the trust of their stakeholders and protect their reputation while benefiting from the efficiency and productivity gains associated with the IIoT.

In summary, the paper's insights into the privacy needs and risks of the IIoT ecosystem are important for policymakers and organizations to understand and develop regulations, guidelines, and strategies that ensure the safety and privacy of the technology and its end-users.

However, in general, some possible limitations or drawbacks of the study include the following:

1. Limited scope or focus: The study may cover only some aspects or perspectives related to the research question, or it may focus on a particular issue, industry, or geography.
2. Methodological limitations: The study may have limitations in data collection, analysis, or interpretation, which can affect the validity and reliability of the findings.
3. Lack of generalizability: The study findings may not be generalizable to other contexts, populations, or settings due to specific sample selection criteria or limitations in the study design.
4. Bias: The study may be affected by biases or assumptions, conscious or unconscious, that can influence the research process, interpretation, and conclusions.

## 10. Conclusions

The IIoT has brought significant benefits to industrial organizations, including greater efficiency, productivity, and cost savings. However, the IIoT also poses significant privacy risks that must be addressed to ensure both the network and its end-users' safety and privacy.

This paper has focused specifically on the privacy needs of the IIoT ecosystem, including protecting personal data and the privacy dimensions specified by industry standards. The paper has also provided an overview of contemporary approaches and solutions for addressing privacy risks in the IIoT, such as encryption and anonymization techniques, access controls and authorization mechanisms, and monitoring and surveillance technologies.

Additionally, the paper has emphasized the need to protect personal data in criminal investigations or the execution of criminal sanctions, particularly in industries subject to regulatory oversight.

Overall, the paper's insights into the privacy needs and risks of the IIoT ecosystem and its recommendations for addressing those risks are important for policymakers and organizations to understand to ensure the safety and privacy of the technology and its end-users. By taking proactive measures to address privacy risks, organizations can maintain the trust of their stakeholders, protect their reputations, and benefit from the efficiency and productivity gains associated with the IIoT.

The main future direction of this research study is a review of blockchain for privacy in the context of the IIoT. It will provide insights into the potential benefits and challenges of using blockchain for privacy in IIoT implementations. It would also help highlight the growing importance of blockchain for enhancing security and privacy in the digital age.

**Author Contributions:** Conceptualization, V.D., S.D. and K.D.; methodology, V.D., S.D. and K.D.; software, V.D., S.D. and K.D.; validation, V.D., S.D. and K.D.; formal analysis, V.D., S.D. and K.D.; investigation, V.D., S.D. and K.D.; resources, V.D., S.D. and K.D.; data curation, V.D., S.D. and K.D.; writing—original draft preparation, V.D., S.D. and K.D.; writing—review and editing, V.D., S.D. and K.D.; visualization, V.D., S.D. and K.D.; supervision, V.D., S.D. and K.D.; project administration, K.D. All authors have read and agreed to the published version of the manuscript.

**Funding:** This research received no external funding.

**Data Availability Statement:** Not applicable.

**Conflicts of Interest:** The authors declare no conflict of interest.

## Abbreviations

| | |
|---|---|
| IIoT | Industrial Internet of Things |
| AI | Artificial Intelligence |
| VR | Virtual Reality |
| QR | Quick Response |
| GDPR | General Data Protection Regulation |
| DPO | Data Protection Officer |
| MAC | Media-Access Control |
| SDN | Software-Defined Network |
| RFID | Radio Frequency Identification |
| ICT | Information and Communication Technology |
| HMI | Human-Machine Interfaces |
| OEM | Original Equipment Manufacturer |
| SHA | Secure Hash Algorithm |
| HMAC | Hash-Based Message Authenticated Code |
| ECDSA | Elliptic Curve Digital Signature Algorithm |
| SIM | Subscriber Identity Module |
| MIM | Machine Identification Module |
| DTLS | Datagram Transport Layer Security |
| MPC | Multiparty Computation |
| LSS | Linear Secret Sharing |
| DPIA | Data Protection Impact Assessments |
| HCI | Human-Computer Interaction |

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
