# Peer review of "An Overview of Privacy Dimensions on the Industrial Internet of Things (IIoT)"

_algorithms, doi:10.3390/a16080378_

Round 1

Reviewer 1 Report (New Reviewer)

The authors reviewed and discussed the privacy needs of an IoT ecosystem , and specified the privacy dimensions, including identity privacy, location privacy, footprint privacy, and multidimensional privacy. The authors also discussed about the potential solution to privacy control, auditability, secure data sharing, especially blockchain technology. In general, this is an interesting work and the privacy topic is also important for the IIOT.  However, the authors needs to address the following concerns before this work being published in this high-quality journal:

  1. With the big context and privacy dimension defined, the author needs to add more discussion about the privacy solutions. Besides blockchain technology, there are also other solutions, for example, federated learning. It would be very helpful if the author could discuss and compare between them;

  2. Different technologies can be applied to different scenarios with its pros and cons. It would be very helpful if the authors could provide some examples for how each technology is applied to different scenarios, For example, for blockchain technology, any existing examples that is utilizing this technology for privacy purposes.

Author Response

Dear respected Reviewer,

We deeply appreciate your time and effort in reviewing our manuscript. Your comments are very helpful for revising and improving our paper. We have revised the manuscript considering all the insightful comments to enhance the paper's readability. We believe these changes have strengthened the rationale and importance of our study.

The authors reviewed and discussed the privacy needs of an IoT ecosystem , and specified the privacy dimensions, including identity privacy, location privacy, footprint privacy, and multidimensional privacy. The authors also discussed about the potential solution to privacy control, auditability, secure data sharing, especially blockchain technology. In general, this is an interesting work and the privacy topic is also important for the IIOT.  However, the authors needs to address the following concerns before this work being published in this high-quality journal:

  1. With the big context and privacy dimension defined, the author needs to add more discussion about the privacy solutions. Besides blockchain technology, there are also other solutions, for example, federated learning. It would be very helpful if the author could discuss and compare between them;

ANS - 1: Thank you for this constructive comment that give us the chance to clarify things further. We have added detailed explanations into introduction section. Specifically, “A discussion of these two approaches (Blockchain and Federated Learning) and comparison presents as follow:

  1. Blockchain Technology: Blockchain technology is a decentralized and distributed ledger system that offers enhanced security and privacy features. It ensures the integrity and immutability of data by storing transactions in a chain of blocks, making it difficult for malicious actors to alter or tamper with the data.

Privacy Benefits:

  1. Data Transparency: Blockchain allows participants in the network to have access to a transparent and auditable history of transactions without revealing specific identifying information.
  2. Data Integrity: The decentralized nature of blockchain ensures that data stored on the ledger is tamper-resistant, making it difficult for unauthorized parties to modify or manipulate information.
  3. Secure Transactions: Blockchain employs cryptographic techniques, such as digital signatures and encryption, to secure data transfers and ensure authenticity.

Challenges:

  1. Scalability: Blockchain networks can face challenges in terms of scalability due to the consensus mechanisms and the need to replicate data across multiple nodes, resulting in slower transaction speeds.
  2. Energy Consumption: Some blockchain networks, particularly those utilizing proof-of-work consensus, require significant computational power, leading to high energy consumption.
  3. Data Privacy: While blockchain technology ensures data integrity and immutability, it does not inherently provide strong privacy protection for the contents of the data. The transparency of blockchain can potentially reveal sensitive information about transactions.
  4. Federated Learning: Federated learning is an approach where machine learning models are trained across multiple decentralized edge devices or servers without sharing the raw data. Instead, only model updates or aggregated information is exchanged between the devices and a central server, ensuring data privacy.

Privacy Benefits:

  1. Data Localization: Federated learning allows data to remain on local devices or servers, reducing the risk of data breaches or unauthorized access.
  2. Privacy-Preserving Model Training: The model updates or aggregated information shared during federated learning are typically anonymized and encrypted, preserving the privacy of individual data points.
  3. Reduced Data Transmission: Federated learning minimizes the need to transfer large amounts of raw data to a central server, which can be beneficial in bandwidth-constrained environments or when dealing with sensitive data.

Challenges:

  1. Model Heterogeneity: Federated learning can be challenging when dealing with a diverse range of edge devices or servers with different computational capabilities, data distributions, or data quality.
  2. Central Server Trust: While federated learning aims to preserve privacy, it still requires trust in the central server that aggregates model updates. A compromised or malicious server could potentially extract sensitive information from the updates.

Comparison:

  1. Data Handling: Blockchain technology stores data directly on the ledger, whereas federated learning keeps the data locally and only exchanges model updates or aggregated information.
  2. Data Privacy: Blockchain provides transparency and integrity but may not provide strong privacy for data contents. Federated learning focuses on preserving the privacy of individual data points during model training.
  3. Trust Model: Blockchain is based on a decentralized trust model, whereas federated learning relies on trust in the central server and the integrity of participants.
  4. Use Cases: Blockchain is suitable for scenarios where data transparency and immutability are critical, such as supply chain management. Federated learning is more applicable when privacy preservation is a priority, such as healthcare or edge device collaboration.

It's important to note that these solutions are not mutually exclusive, and their applicability depends on the specific requirements and constraints of the IIoT ecosystem. Organizations may choose to combine these approaches or utilize other privacy-enhancing technologies to achieve the desired level of privacy and security.”

  1. Different technologies can be applied to different scenarios with its pros and cons. It would be very helpful if the authors could provide some examples for how each technology is applied to different scenarios, For example, for blockchain technology, any existing examples that is utilizing this technology for privacy purposes.

ANS - 2: Thank you for this helpful comment. We have added detailed explanations into introduction section. Specifically, “Here are a few examples of how blockchain technology has been applied in various scenarios to enhance privacy:

  1. Healthcare Data Sharing: Blockchain can be used to improve the privacy and security of healthcare data sharing. By storing medical records and sensitive patient information on a blockchain, access can be controlled, and data integrity can be ensured. Patients have control over their own data and can grant access to healthcare providers as needed, reducing the risk of unauthorized access or data breaches. One example is MedRec, a blockchain-based system that allows patients to securely share their medical records with healthcare providers while maintaining privacy and data ownership.
  2. Supply Chain Management: Blockchain technology has found applications in enhancing privacy and transparency in supply chain management. By recording transactions and tracking products on a blockchain, stakeholders can verify the authenticity and provenance of goods without revealing sensitive business information. This helps prevent counterfeit products and provides transparency for consumers. IBM's Food Trust is a notable example that utilizes blockchain to track and trace food products, ensuring the integrity of the supply chain and providing consumers with information about the origin and handling of their food.
  3. Identity Management: Blockchain offers potential solutions for secure and privacy-preserving identity management systems. By using blockchain, individuals can maintain control over their personal data and selectively disclose information to third parties, reducing the risk of identity theft and unauthorized data access. Self-sovereign identity (SSI) solutions, such as uPort and Sovrin, leverage blockchain to enable individuals to manage and control their digital identities, providing privacy-enhancing features and reducing reliance on centralized identity providers.
  4. Financial Transactions and Privacy: Blockchain technology can improve privacy in financial transactions by reducing the need for trusted intermediaries and providing pseudonymity. Cryptocurrencies like Bitcoin and privacy-focused cryptocurrencies like Monero utilize blockchain to facilitate secure, decentralized, and pseudonymous transactions. While blockchain transactions are public, privacy-focused cryptocurrencies employ techniques such as ring signatures, stealth addresses, and zero-knowledge proofs to obfuscate transaction details and enhance privacy.

It's worth noting that while blockchain technology can enhance privacy in these scenarios, its implementation requires careful consideration of the specific use case, including factors such as regulatory compliance, scalability, and user adoption. Blockchain is not a one-size-fits-all solution, and organizations need to evaluate its feasibility and benefits for their specific privacy requirements.

Here are a few examples of how federated learning technology has been applied in different scenarios to enhance privacy:

  1. Healthcare: Federated learning can be applied in healthcare settings to enable collaborative model training while preserving patient privacy. Hospitals or medical institutions can train machine learning models using local patient data without sharing sensitive patient information. The models are then aggregated or updated in a privacy-preserving manner, allowing healthcare providers to benefit from shared insights without compromising patient confidentiality. This approach can be useful for applications such as disease diagnosis, medical image analysis, or predictive analytics.
  2. Smart Devices and Internet of Things (IoT): Federated learning is well-suited for scenarios involving edge devices or IoT devices. These devices often have limited computational resources, making it challenging to send large amounts of data to a centralized server for model training. With federated learning, edge devices can collaboratively train machine learning models using locally collected data while keeping the data on the device. Only model updates or aggregated information is sent to a central server, ensuring privacy while benefiting from shared knowledge. This is useful in applications such as personalized recommendations, activity recognition, or anomaly detection in smart homes or industrial IoT settings.
  3. Financial Services: Federated learning can enhance privacy in financial services by enabling collaborative model training while keeping sensitive customer data on local servers or devices. Banks or financial institutions can train machine learning models for tasks like fraud detection or credit scoring using locally held customer data. The models' updates or aggregated information are exchanged in a privacy-preserving manner, ensuring the privacy of individual customer transactions and sensitive financial information.
  4. Natural Language Processing (NLP): Federated learning can be applied in NLP tasks to protect user privacy while improving language models. Instead of centralizing user data on a single server, federated learning allows individual devices or servers to train language models using local data. The models' updates or aggregated information, which preserve the privacy of individual texts, are shared across devices or servers. This approach enhances privacy while enabling the improvement of language models for applications such as voice assistants, chatbots, or sentiment analysis.

These examples illustrate how federated learning can be leveraged in various domains to enable collaborative model training while maintaining privacy and data confidentiality. By keeping data decentralized and only exchanging model updates or aggregated information, federated learning offers a privacy-enhancing approach for machine learning in scenarios where data privacy is crucial.

Reviewer 2 Report (New Reviewer)

This manuscript attempts to provide the review of existing privacy issues in the IIoT domain, study the associated threats and outline the relevant recommendations.
However, the work at hand has a number of significant flows.

1. The scientific value of the manuscript and followed methodology is not defined.
In particular, what are the research questions? How this work is different from the other review papers? What are the contributions of the authors, apart from providing high-level review?
2. Findings presentations is very poor.
First, the language needs major improvement from grammatical point of view and also how the sentences are composed. It is very difficult to follow  the logic and story line of the manuscript. Many statements are either not reasoned or no references are provided or statements are simply misleading. The content is overlapping between sections and within sections.

It is highly recommended for the authors to rethink what exactly they intend to demonstrate within this manuscript and think about a continuous story with decent explanation of newly appearing terms and their relevance to the discussion.

More detailed remarks below:
- lines 39-40: meaning unclear "In the future, almost every manufactured item - whether infrastructure or consumable - may be embedded with sensors," --> how these two categories are mutually exclusive?
- lines 48-49: suddenly jumping into blockchain. no introduction, no reasoning. self-citation, but previous work is not explained nor reasoned.
- lines 55-58: link with UAVs as an example is not clear. why it is "most important", why not lidar equipped vehicles, why not satellites, why not sensors installed throughout the city?
- how AI helps to use data in real-time?
- lines 71-79: reasoning is not clear. what is "local socket"? how 5G makes company's network "isolated"?
- lines 88-89: how "machine data" can improve "customer relationships"?
- Introduction: how point 1 is different from point 4, both are describing machine learning.
- The purpose of figure 3 is unclear (self-citation again) and does not reflect promised "privacy and security IIoT scenario".
- Contributions: how 2 is different from 3 and 5? 3 and 5 identically to 2, state investigation into "privacy needs". Finally, only 3 contributions are described: overview of AI use in IIoT, investigation into security and privacy needs, overview of the existing approaches and solutions
- section 2: lines 237-252: how these are "sectors"? especially DPO appointment? Why industrial domain should have a specific privacy regulation and general existing regulations such as GDPR are not relevant?
- the link between privacy dimensions presented first (lines 206-230) and later (258-270) is unclear. How they are related? The second set seems to be generic security properties coming from CIA triad combined with "Data protection".
- section 3: what is the purpose of this section? the reasoning is very weak for the definition of industrial privacy concept, which is no different from the privacy needs in any other domain (at least no such arguments were presented by the authors).   
- section 4: lines 366-367: how microprocessor can be IP-based?!
- lines 367-369: reasoning? references? boosting what? how?
- lines 369-371: how ICS or HMI can be an example of an attack?
- lines 373-377: grouped where? how? reference?
- what is the purpose of figure 4? how is it linked with this section?
- lines 446-544: how the title under numbered list are "threats"? (eg Identification and Authorization). What is threat under number 3?
- how figure 5 is related to threats discussion (once again, self-citation).
- why there is a second numbered list of "most common privacy threats in the IIoT"? the previous one was not enough? No referencing is provided.
- lines 634-673: how blockchain is relevant here?
- how data can be transparent and private at the same time?
- lines 677-680: why listing privacy requirements again here? and why are they different from previous section?
- section 6: limitations. are those your limitations? or is it a generic description of possible limitations that can be applicable to any kind of research?
- Abbreviations should be normalised and used properly throughout manuscript. For example, A.I --> AI; V.R --> VR; what is R.V.R?

- lines 46-47: "which upgrade substantial the industrial environment [4]." --> substantially
- lines 276-278: meaning of the sentence is unclear. unfinished sentence.

Author Response

Dear respected Reviewer,

We deeply appreciate your time and effort in reviewing our manuscript. Your comments are very helpful for revising and improving our paper. We have revised the manuscript considering all the insightful comments to enhance the paper's readability. We believe these changes have strengthened the rationale and importance of our study.

This manuscript attempts to provide the review of existing privacy issues in the IIoT domain, study the associated threats and outline the relevant recommendations.
However, the work at hand has a number of significant flows.

1. The scientific value of the manuscript and followed methodology is not defined.
In particular, what are the research questions? How this work is different from the other review papers? What are the contributions of the authors, apart from providing high-level review?

ANS – 2. Thank you for this constructive comment. We have added detailed explanations into introduction section. Specifically, “The study investigate the privacy needs of the IIoT ecosystem, particularly in relation to the processing of personal data by competent authorities for crime prevention, investigation, detection, prosecution, or execution of criminal sanctions. The study also provides an overview of contemporary approaches and solutions for addressing industrial privacy risks and concludes by examining privacy needs and recommendations for an ideal safe and private IIoT ecosystem.

This work, have differentiated from other review papers in the following ways:

  1. Original Research or Insights: The research study conducted original research and provided new insights into the privacy needs and challenges specific to the industrial environment in the context of IIoT. This involve analyzing existing frameworks, identifying gaps, proposing novel solutions, and providing practical recommendations tailored to the industrial sector.
  2. Industry-specific Perspective: The paper have focused on the unique privacy requirements and considerations specific to industrial organizations. This involve examining privacy concerns in industrial settings, addressing the challenges of heterogeneous and complex IIoT systems, and discussing privacy implications within specific industries such as manufacturing, energy, or healthcare.
  3. Evaluation of Contemporary Approaches: The study have critically evaluated existing approaches and solutions for addressing privacy risks in industrial environments. This include assessing the effectiveness, scalability, and practicality of different techniques, frameworks, and technologies in ensuring privacy within IIoT deployments.
  4. Practical Recommendations: Apart from providing a high-level review, the paper have offered practical recommendations for establishing a safe and private IIoT ecosystem in industrial settings. These recommendations involve specific guidelines, best practices, and implementation strategies to enhance privacy protection and mitigate risks.

  5. Findings presentations is very poor.

ANS – 2. Thank you for your helpful comment. The discussion section was rearranged based on your suggestions.  

First, the language needs major improvement from grammatical point of view and also how the sentences are composed. It is very difficult to follow  the logic and story line of the manuscript. Many statements are either not reasoned or no references are provided or statements are simply misleading. The content is overlapping between sections and within sections. It is highly recommended for the authors to rethink what exactly they intend to demonstrate within this manuscript and think about a continuous story with decent explanation of newly appearing terms and their relevance to the discussion.
More detailed remarks below:
- lines 39-40: meaning unclear "In the future, almost every manufactured item - whether infrastructure or consumable - may be embedded with sensors," --> how these two categories are mutually exclusive?

ANS. Thank you for your remarks. The text was rearranged based on your suggestions.  

- lines 48-49: suddenly jumping into blockchain. no introduction, no reasoning. self-citation, but previous work is not explained nor reasoned.

ANS. Thank you for your remarks. The manuscript was rearranged based on your suggestions.  

- lines 55-58: link with UAVs as an example is not clear. why it is "most important", why not lidar equipped vehicles, why not satellites, why not sensors installed throughout the city?

ANS. Thank you for your remarks. The text was rearranged based on your suggestions. Specifically, “Unmanned Aerial Vehicles (Drones, UAVs) are one of the most important information-gathering systems”.  

- how AI helps to use data in real-time?

ANS. Thank you for your remarks. As mentioned in the revised manuscript, “AI plays a crucial role in leveraging data in real-time by enabling efficient processing, analysis, and decision-making. Here are several ways AI helps in utilizing data in real-time:

  1. Real-time Data Processing: AI algorithms and techniques are employed to process large volumes of data in real-time. This includes tasks such as data filtering, aggregation, transformation, and normalization, which are essential for extracting valuable insights from the data as it flows in.
  2. Real-time Analytics: AI-powered analytics tools can analyze data streams in real-time to identify patterns, trends, anomalies, or correlations. These real-time analytics capabilities allow organizations to gain immediate insights and make data-driven decisions as new data is generated.
  3. Predictive Analytics: AI algorithms, particularly machine learning and deep learning models, can be trained to analyze real-time data and make predictions or forecasts. This enables organizations to anticipate events or outcomes based on the most recent data, facilitating proactive decision-making.
  4. Intelligent Automation: AI techniques, such as natural language processing (NLP) and computer vision, enable the automation of data processing tasks in real-time. This includes automated data extraction, classification, and summarization, which accelerates the analysis and utilization of real-time data.
  5. Real-time Decision Support: AI systems can provide real-time recommendations or suggestions based on the analysis of incoming data. These recommendations can aid in decision-making processes by providing insights, identifying opportunities, or suggesting actions in real-time.
  6. Fraud Detection and Anomaly Detection: AI algorithms can continuously analyze data streams to detect fraudulent activities or anomalies in real-time. This is particularly useful in financial transactions, cybersecurity, or industrial monitoring, where prompt identification of anomalies is critical.
  7. Personalized Experiences: AI-powered systems can leverage real-time data to deliver personalized experiences to users or customers. By analyzing user behavior, preferences, or contextual data in real-time, AI algorithms can tailor recommendations, content, or services in a personalized and timely manner.
  8. Real-time Insights Integration: AI systems can integrate real-time insights with existing business processes, systems, or applications. This allows organizations to incorporate real-time data-driven decision-making directly into their operational workflows, enhancing agility and responsiveness.

AI empowers organizations to harness the power of real-time data by enabling efficient processing, analysis, and decision-making. By leveraging AI algorithms and techniques, organizations can extract insights, make predictions, automate processes, and provide personalized experiences in real-time, thereby gaining a competitive edge and driving innovation.”

- lines 71-79: reasoning is not clear. what is "local socket"? how 5G makes company's network "isolated"?

ANS. Thank you for your feedback. We have added explanations in the revised manuscript, Specifically, “"Local socket" refers to a communication endpoint or interface that allows processes running on the same device or within the same local network to communicate with each other. It enables interprocess communication, facilitating data exchange and coordination between different software components or applications within a device or local network. In the context of networking, a local socket typically refers to a network socket that is bound to a specific IP address and port on a device within a local network. It allows processes or applications on the same device or local network to establish connections and exchange data using standard networking protocols.

Regarding 5G and network isolation, 5G networks introduce certain features and technologies that enhance network isolation and security for organizations. For example, Network Slicing, Enhanced Security Mechanisms, Private 5G Networks and Network Function Virtualization (NFV). By leveraging these 5G capabilities, organizations can create isolated network environments that enhance security, privacy, and control. These isolated networks help protect sensitive data, enable secure communications, and provide dedicated resources tailored to specific organizational needs.”
- lines 88-89: how "machine data" can improve "customer relationships"?

ANS. Thank you for your remarks. We have added explanations in the revised manuscript, Specifically, “The statement highlights the impact of interconnected devices and the data they generate on enhancing customer relationships. Specifically, connected machines and the data they generate enable organizations to leverage machine data in various ways, leading to improved customer relationships. By utilizing machine data for proactive maintenance, predictive analytics, customized offerings, and enhanced support, organizations can deliver better customer experiences, increase customer satisfaction, and strengthen their relationships with customers.”
- Introduction: how point 1 is different from point 4, both are describing machine learning.

ANS. Thank you for your remarks. We have added explanations in the revised manuscript, Specifically, “Real-time data processing focuses on the timely processing and analysis of data as it is generated, enabling organizations to make immediate decisions or take prompt actions. Intelligent automation, on the other hand, leverages AI and automation technologies to automate tasks and processes, reducing human effort and improving efficiency. Together, they can enhance operational agility, optimize decision-making, and drive intelligent actions based on real-time data.”
- The purpose of figure 3 is unclear (self-citation again) and does not reflect promised "privacy and security IIoT scenario".

ANS. Thank you for your remarks. Figure 3 presents the Federated Auto-Meta-Ensemble Learning (FAMEL) architecture in the new IT/OT Industrial environment. This is the second round of review for the paper. We decided to use pertinent figures from our papers after one of the first round's reviewers emphasized that all figures needed copyrights.
- Contributions: how 2 is different from 3 and 5? 3 and 5 identically to 2, state investigation into "privacy needs". Finally, only 3 contributions are described: overview of AI use in IIoT, investigation into security and privacy needs, overview of the existing approaches and solutions

ANS. Thank you for your remarks. We have revised the manuscript, as follow:

  1. Identification of Privacy Dimensions: The study specifies the privacy dimensions required for protecting individuals within the IoT ecosystem, particularly in relation to the processing of personal data by competent authorities for crime prevention, investigation, detection, prosecution, or execution of criminal sanctions. By clearly defining these privacy dimensions, the study provides a foundation for developing effective privacy measures in IIoT systems.
  2. Overview of Contemporary Approaches: The study offers an overview of contemporary approaches and solutions for mitigating privacy risks in industrial settings. It explores the existing techniques, technologies, and best practices employed to ensure data privacy and security in IIoT systems. This overview serves as a valuable resource for organizations seeking to enhance privacy protection in their IIoT implementations.
  3. Recommendations for a Safe and Private IIoT Ecosystem: Based on the findings and insights gathered from the investigation, the study concludes by providing recommendations for establishing an ideal, safe, and private IIoT ecosystem. These recommendations offer practical guidance for organizations to implement privacy-enhancing measures, build trust, and ensure compliance with industry standards.

- section 2: lines 237-252: how these are "sectors"? especially DPO appointment? Why industrial domain should have a specific privacy regulation and general existing regulations such as GDPR are not relevant?

ANS. Thank you for your remarks. The text was rearranged based on your suggestions. Specifically, “The industrial domain often requires specific privacy regulations due to its unique characteristics and privacy concerns that may not be fully addressed by general existing regulations like the General Data Protection Regulation (GDPR). Here are some reasons why the industrial domain may benefit from specific privacy regulations:

  1. Specialized Data Types: The industrial domain deals with specific types of data that may not be explicitly covered by general regulations. Industrial environments often involve sensitive data related to manufacturing processes, proprietary technologies, trade secrets, industrial control systems, or safety protocols. These data types require specialized privacy considerations to protect intellectual property, ensure operational safety, and prevent unauthorized access or misuse.
  2. Complex Data Ecosystems: Industrial environments typically have complex data ecosystems with interconnected machines, sensors, and control systems. These systems generate and exchange vast amounts of data, often in real-time. General regulations like GDPR may not adequately address the intricacies of managing and securing data within such heterogeneous and dynamic environments. Specific regulations can provide guidelines and requirements tailored to the unique challenges of industrial data ecosystems.
  3. Safety and Security Risks: Privacy regulations in the industrial domain need to consider not only the protection of personal data but also the safety and security risks associated with industrial processes. Data breaches or unauthorized access in industrial settings can have severe consequences, including physical harm, environmental damage, or disruptions to critical infrastructure. Specific regulations can address these risks and impose safeguards to mitigate potential threats.
  4. Industry-Specific Compliance Requirements: Different industries within the industrial domain, such as manufacturing, energy, or healthcare, may have specific compliance requirements related to privacy and data protection. These requirements may be driven by sector-specific regulations, standards, or contractual obligations. Specific privacy regulations can align with these industry-specific compliance requirements to ensure that organizations within the industrial domain adhere to the necessary privacy practices.
  5. Operational Constraints and Challenges: The industrial domain often faces operational constraints and challenges that are distinct from other sectors. These may include limited connectivity, remote locations, harsh environments, or legacy systems. General privacy regulations may not consider these operational constraints, making it necessary to have specific regulations that accommodate the unique circumstances of the industrial domain while still ensuring data privacy.

It's important to note that while general regulations like GDPR provide a valuable framework for data protection and privacy, the industrial domain may require additional measures and specific regulations to address its unique characteristics, data types, ecosystem complexity, safety risks, compliance requirements, and operational challenges. The specific regulations can build upon and complement general regulations, providing a more comprehensive and tailored approach to privacy in the industrial context.”

- the link between privacy dimensions presented first (lines 206-230) and later (258-270) is unclear. How they are related? The second set seems to be generic security properties coming from CIA triad combined with "Data protection".

ANS. Thank you for your comment. Privacy dimensions refer to different aspects or elements of privacy that are considered when designing privacy protection mechanisms or frameworks. These dimensions encompass various facets of privacy that need to be addressed to ensure the confidentiality, integrity, and control of personal information. These privacy dimensions collectively provide a comprehensive framework for assessing and addressing privacy concerns. Organizations and policymakers consider these dimensions to develop privacy policies, establish privacy controls, and design privacy-preserving systems and practices. By addressing each dimension, organizations can work towards creating a privacy-respecting environment that safeguards personal data and respects individuals' privacy rights.

- section 3: what is the purpose of this section? the reasoning is very weak for the definition of industrial privacy concept, which is no different from the privacy needs in any other domain (at least no such arguments were presented by the authors).  

 ANS. Thank you for your remarks. The text was rearranged based on your suggestions. Specifically, “The industrial domain often requires specific privacy regulations due to its unique characteristics and privacy concerns that may not be fully addressed by general existing regulations like the General Data Protection Regulation (GDPR). Here are some reasons why the industrial domain may benefit from specific privacy regulations:

  1. Specialized Data Types: The industrial domain deals with specific types of data that may not be explicitly covered by general regulations. Industrial environments often involve sensitive data related to manufacturing processes, proprietary technologies, trade secrets, industrial control systems, or safety protocols. These data types require specialized privacy considerations to protect intellectual property, ensure operational safety, and prevent unauthorized access or misuse.
  2. Complex Data Ecosystems: Industrial environments typically have complex data ecosystems with interconnected machines, sensors, and control systems. These systems generate and exchange vast amounts of data, often in real-time. General regulations like GDPR may not adequately address the intricacies of managing and securing data within such heterogeneous and dynamic environments. Specific regulations can provide guidelines and requirements tailored to the unique challenges of industrial data ecosystems.
  3. Safety and Security Risks: Privacy regulations in the industrial domain need to consider not only the protection of personal data but also the safety and security risks associated with industrial processes. Data breaches or unauthorized access in industrial settings can have severe consequences, including physical harm, environmental damage, or disruptions to critical infrastructure. Specific regulations can address these risks and impose safeguards to mitigate potential threats.
  4. Industry-Specific Compliance Requirements: Different industries within the industrial domain, such as manufacturing, energy, or healthcare, may have specific compliance requirements related to privacy and data protection. These requirements may be driven by sector-specific regulations, standards, or contractual obligations. Specific privacy regulations can align with these industry-specific compliance requirements to ensure that organizations within the industrial domain adhere to the necessary privacy practices.
  5. Operational Constraints and Challenges: The industrial domain often faces operational constraints and challenges that are distinct from other sectors. These may include limited connectivity, remote locations, harsh environments, or legacy systems. General privacy regulations may not consider these operational constraints, making it necessary to have specific regulations that accommodate the unique circumstances of the industrial domain while still ensuring data privacy.

It's important to note that while general regulations like GDPR provide a valuable framework for data protection and privacy, the industrial domain may require additional measures and specific regulations to address its unique characteristics, data types, ecosystem complexity, safety risks, compliance requirements, and operational challenges. The specific regulations can build upon and complement general regulations, providing a more comprehensive and tailored approach to privacy in the industrial context.”
- section 4: lines 366-367: how microprocessor can be IP-based?!

ANS. Thank you for your remarks. A microprocessor can be IP-based when it incorporates intellectual property (IP) cores or modules into its design. An IP core refers to a pre-designed and reusable component that can be integrated into a larger system or microprocessor design. These IP cores are typically developed by third-party vendors or IP providers and offer specific functionalities or features.

- lines 367-369: reasoning? references? boosting what? how?

 ANS. Thank you for your remarks. The text was rearranged based on your suggestions. Specifically, “IIoT solutions can improve productivity by automating tasks, streamlining processes, and enabling efficient resource management. Connected devices and sensors can collect data on equipment performance, inventory levels, and production metrics, allowing organizations to identify bottlenecks, optimize workflows, and reduce downtime. This leads to improved operational efficiency, reduced costs, and increased productivity. (https://doi.org/10.1016/j.procs.2021.12.179)”
- lines 369-371: how ICS or HMI can be an example of an attack?

 ANS. Thank you for your remarks. The text was rearranged based on your suggestions. Specifically, “ICS and HMI can be vulnerable to various types of attacks, which can have severe consequences on industrial operations. For example:

  1. Malware and Ransomware: Attackers can deploy malware or ransomware specifically designed to target ICS or HMI systems. These malicious programs can disrupt the functioning of critical industrial processes, compromise system integrity, and even demand ransom payments in exchange for restoring normal operations.
  2. Distributed Denial of Service (DDoS): A DDoS attack involves overwhelming a system with a flood of traffic, causing it to become unresponsive or crash. If an attacker successfully launches a DDoS attack against an ICS or HMI system, it can disrupt control signals, delay response times, or render the system inoperable, leading to production interruptions or safety risks.
  3. Unauthorized Access and Control: If an attacker gains unauthorized access to an ICS or HMI system, they can manipulate control parameters, change setpoints, or issue unauthorized commands. This can lead to process deviations, equipment damage, safety hazards, or even catastrophic incidents.
  4. Insider Threats: Insiders with malicious intent, such as disgruntled employees or contractors, can abuse their privileged access to ICS or HMI systems. They may deliberately tamper with control settings, sabotage equipment, or steal sensitive data, causing significant disruptions or compromising system integrity.
  5. Social Engineering: Attackers may employ social engineering techniques to trick authorized users into divulging sensitive information or granting unauthorized access. For example, phishing emails or phone calls can deceive employees into revealing login credentials or executing malicious commands, which can be used to compromise ICS or HMI systems.
  6. Supply Chain Attacks: ICS or HMI systems can be targeted through vulnerabilities in the supply chain. Attackers may compromise the integrity of hardware, software, or firmware during the manufacturing, distribution, or installation process. This can result in the introduction of malicious components or exploitable weaknesses in the system.

The consequences of these attacks can range from operational disruptions and financial losses to safety incidents and environmental hazards. Protecting ICS and HMI systems requires robust security measures, including network segmentation, regular software updates and patches, strong access controls, user training, intrusion detection systems, and incident response plans.

It's important to note that the examples provided are not exhaustive, and the specific attack vectors and techniques can vary depending on the specific ICS or HMI system in use. Organizations need to assess and mitigate risks based on their unique operational requirements and the potential threats they face.
- lines 373-377: grouped where? how? reference?

 ANS. Thank you for your remarks. The text was rearranged based on your suggestions. Specifically, “Grouping vulnerabilities by categories is a useful approach to organize and address the wide range of vulnerabilities that can affect Information Technology (IT) and Operational Technology (OT) systems in an Industrial Control System (ICS) environment. Vulnerabilities can be grouped as Policy Concerns, Practice Concerns, Hardware Vulnerabilities, Operating System Vulnerabilities, ICS Application Vulnerabilities, etc. These categories can help in establishing mitigation measures and prioritizing security efforts. (urn:nbn:se:kth:diva-302147)”
- what is the purpose of figure 4? how is it linked with this section?

 ANS. Thank you for your remarks. The Blockchained AutoML Network Traffic Analyzer (BANTA) system uses smart contracts to automate the process of network traffic analysis and store network traffic logs in a decentralized manner, providing transparency and security in the ICS.
- lines 446-544: how the title under numbered list are "threats"? (eg Identification and Authorization). What is threat under number 3?

 ANS. Thank you for your careful reading. In the revised manuscript, the numbered list is “potential risks”. Number 3 was the typing fault. The text was rearranged.
- how figure 5 is related to threats discussion (once again, self-citation).

 ANS. Thank you for this comment. That gives us the chance to clarify things further. The figure 5 is a Privacy-preserving Industrial Framework that includes 10 modules. Is an abstract and high-level depiction of the specific industrial privacy-preserving framework.
- why there is a second numbered list of "most common privacy threats in the IIoT"? the previous one was not enough? No referencing is provided.

 ANS. Thank you for your remarks. The text was rearranged based on your suggestions. Specifically, “In light of those above, the IIoT presents unique privacy challenges and threats due to the large amounts of data generated, stored, and processed in industrial environments. Some of the most common privacy threats in the IIoT include Data Breaches, Cyberattacks, Insider Threats, Lack of Transparency and Unsecured Communications.

To address these privacy threats, IIoT implementations must implement appropriate technical and organizational measures to protect sensitive information and personal data. These measures may include encryption, access controls, employee training, regular vul-nerability assessments, and ongoing IIoT systems and devices monitoring. By imple-menting these measures, IIoT systems can better protect against privacy threats and en-sure sensitive information's confidentiality, integrity, and availability.”

- lines 634-673: how blockchain is relevant here?

ANS. Thank you for this constructive comment that give us the chance to clarify things further. We have added detailed explanations into introduction section. Specifically, “A discussion of these two approaches (Blockchain and Federated Learning) and comparison presents as follow:

  1. Blockchain Technology: Blockchain technology is a decentralized and distributed ledger system that offers enhanced security and privacy features. It ensures the integrity and immutability of data by storing transactions in a chain of blocks, making it difficult for malicious actors to alter or tamper with the data.

Privacy Benefits:

  1. Data Transparency: Blockchain allows participants in the network to have access to a transparent and auditable history of transactions without revealing specific identifying information.
  2. Data Integrity: The decentralized nature of blockchain ensures that data stored on the ledger is tamper-resistant, making it difficult for unauthorized parties to modify or manipulate information.
  3. Secure Transactions: Blockchain employs cryptographic techniques, such as digital signatures and encryption, to secure data transfers and ensure authenticity.

Challenges:

  1. Scalability: Blockchain networks can face challenges in terms of scalability due to the consensus mechanisms and the need to replicate data across multiple nodes, resulting in slower transaction speeds.
  2. Energy Consumption: Some blockchain networks, particularly those utilizing proof-of-work consensus, require significant computational power, leading to high energy consumption.
  3. Data Privacy: While blockchain technology ensures data integrity and immutability, it does not inherently provide strong privacy protection for the contents of the data. The transparency of blockchain can potentially reveal sensitive information about transactions.
  4. Federated Learning: Federated learning is an approach where machine learning models are trained across multiple decentralized edge devices or servers without sharing the raw data. Instead, only model updates or aggregated information is exchanged between the devices and a central server, ensuring data privacy.

Privacy Benefits:

  1. Data Localization: Federated learning allows data to remain on local devices or servers, reducing the risk of data breaches or unauthorized access.
  2. Privacy-Preserving Model Training: The model updates or aggregated information shared during federated learning are typically anonymized and encrypted, preserving the privacy of individual data points.
  3. Reduced Data Transmission: Federated learning minimizes the need to transfer large amounts of raw data to a central server, which can be beneficial in bandwidth-constrained environments or when dealing with sensitive data.

Challenges:

  1. Model Heterogeneity: Federated learning can be challenging when dealing with a diverse range of edge devices or servers with different computational capabilities, data distributions, or data quality.
  2. Central Server Trust: While federated learning aims to preserve privacy, it still requires trust in the central server that aggregates model updates. A compromised or malicious server could potentially extract sensitive information from the updates.

Comparison:

  1. Data Handling: Blockchain technology stores data directly on the ledger, whereas federated learning keeps the data locally and only exchanges model updates or aggregated information.
  2. Data Privacy: Blockchain provides transparency and integrity but may not provide strong privacy for data contents. Federated learning focuses on preserving the privacy of individual data points during model training.
  3. Trust Model: Blockchain is based on a decentralized trust model, whereas federated learning relies on trust in the central server and the integrity of participants.
  4. Use Cases: Blockchain is suitable for scenarios where data transparency and immutability are critical, such as supply chain management. Federated learning is more applicable when privacy preservation is a priority, such as healthcare or edge device collaboration.

It's important to note that these solutions are not mutually exclusive, and their applicability depends on the specific requirements and constraints of the IIoT ecosystem. Organizations may choose to combine these approaches or utilize other privacy-enhancing technologies to achieve the desired level of privacy and security.”
- how data can be transparent and private at the same time?

 ANS. Thank you for your remarks. Ensuring data transparency and privacy can be challenging but not impossible. While they may seem contradictory, it is possible to strike a balance between data transparency and privacy through careful design and implementation of data management practices. Here's how it can be achieved:

  1. Data Governance and Consent: Implementing robust data governance practices is essential. Clearly define data collection, usage, and sharing policies, and obtain explicit consent from individuals whose data is being collected. Transparently communicate how the data will be used, who will have access to it, and for what purposes.
  2. Anonymization and Aggregation: To protect privacy, sensitive data can be anonymized or aggregated. Anonymization involves removing or de-identifying personally identifiable information (PII) from the dataset, making it difficult to link specific individuals to the data. Aggregation combines data from multiple sources to provide insights without revealing individual-level information.
  3. Access Controls and Encryption: Implement strong access controls to restrict data access to authorized personnel only. Encryption techniques can be employed to secure data at rest and in transit, ensuring that even if the data is accessed, it remains protected and unreadable to unauthorized parties.
  4. Purpose Limitation and Data Minimization: Collect and retain only the data that is necessary for the intended purpose. Avoid unnecessary data collection and ensure that data is only used for the specific purposes for which consent was obtained.
  5. Transparency in Data Handling: Clearly communicate to individuals how their data is handled, including storage, processing, and any third-party involvement. Provide transparency reports that outline the data management practices, data security measures, and compliance with privacy regulations.
  6. User Control and Consent Management: Give individuals control over their data through user-centric tools and interfaces. Enable them to easily access, review, update, and delete their data, as well as manage their consent preferences.
  7. Regular Auditing and Compliance: Conduct regular audits and assessments to ensure compliance with privacy regulations and internal policies. Regularly review and update data management practices to adapt to changing privacy requirements and emerging risks.
  8. Privacy by Design: Embed privacy considerations into the design and development of systems and processes from the outset. Consider privacy as a fundamental requirement rather than an afterthought, implementing privacy-enhancing technologies and practices throughout the data lifecycle.

It's important to note that achieving a balance between data transparency and privacy requires a multidimensional approach that considers legal, technical, and ethical aspects. Organizations must comply with applicable privacy laws and regulations while also respecting individual privacy rights and maintaining transparency in their data practices.

By implementing these practices, organizations can maintain a level of transparency regarding data collection, usage, and sharing, while also safeguarding individuals' privacy by protecting their sensitive information and ensuring appropriate data handling practices.

- lines 677-680: why listing privacy requirements again here? and why are they different from previous section?

 ANS. Thank you for your remarks. The privacy requirements are listed for the first time in this section.
- section 6: limitations. are those your limitations? or is it a generic description of possible limitations that can be applicable to any kind of research?

 ANS. Thank you for your suggestions. In the revised section 7 listed some possible limitations or drawbacks of the study that also can be applicable to any kind of research.
- Abbreviations should be normalised and used properly throughout manuscript. For example, A.I --> AI; V.R --> VR; what is R.V.R?

ANS. Thank you for your carefully reading. The paper was improved based on your suggestions.

Comments on the Quality of English Language

- lines 46-47: "which upgrade substantial the industrial environment [4]." --> substantially
- lines 276-278: meaning of the sentence is unclear. unfinished sentence.

ANS. Thank you for your carefully reading. The quality of English language was improved based on your suggestions.  

Reviewer 3 Report (New Reviewer)

The paper presents a survey of privacy issues in IIOT. Overall, the paper is well written with simplicity in mind and conveys successfully the presented notions to the reader. I have some issues only in the presentation of the paper in some parts of it:

I was not able to understand why the figure of an IoT architecture called advocate was included in the paper. What is the motivation of this selection and not of other solutions?

I would suggest to move down the tables in the introduction and in general reduce the size of the introduction. Moreover, some of the contributions of the paper should be merged to two or three and not five as they are very simple and somewhat repetitive. 

I was not able to understand why figure 5 was included in the paper. It seems to be a set of modules connected to each other but really does not add anything useful in the paper.

Can you please explain what is AutoML and why you are presenting it?

I would suggest to remove the sentence "Incomplete or outdated information: The study may include something other than the most recent or relevant information, data, or literature related to the research question of blockchain technology." Why this may happen? Why only to blockchain technology? 

Could you please add some references for GDPR and IIoT. 

Figure 1 has a typo (bossines)

Page 398 (repetition of word "the")

Some minor comments as written above.

Author Response

Dear respected Reviewer,

We deeply appreciate your time and effort in reviewing our manuscript. Your comments are very helpful for revising and improving our paper. We have revised the manuscript considering all the insightful comments to enhance the paper's readability. We believe these changes have strengthened the rationale and importance of our study.

The paper presents a survey of privacy issues in IIOT. Overall, the paper is well written with simplicity in mind and conveys successfully the presented notions to the reader. I have some issues only in the presentation of the paper in some parts of it:

  1. I was not able to understand why the figure of an IoT architecture called advocate was included in the paper. What is the motivation of this selection and not of other solutions?

ANS – 1. Thank you for your feedback. The Advocate is an example architecture that promotes privacy and security in the IoT/IIoT ecosystem.

  1. I would suggest to move down the tables in the introduction and in general reduce the size of the introduction. Moreover, some of the contributions of the paper should be merged to two or three and not five as they are very simple and somewhat repetitive. 

ANS – 2. Thank you for this constructive comment. The introduction was rearranged based on your suggestions.  

  1. I was not able to understand why figure 5 was included in the paper. It seems to be a set of modules connected to each other but really does not add anything useful in the paper.

ANS – 3. Thank you for this comment. That gives us the chance to clarify things further. The figure 5 is a Privacy-preserving Industrial Framework that includes 10 modules. Is an abstract and high-level depiction of the specific industrial privacy-preserving framework.

  1. Can you please explain what is AutoML and why you are presenting it?

ANS – 4. Thank you for your carefully reading. The figure 3 is a depiction of the Federated Auto-Meta-Ensemble Learning (FAMEL) architecture. It is a holistic system that automates selecting and using the most appropriate algorithmic hyperparameters that optimally solve a problem under consideration, approaching it as a model for finding algorithmic solutions where it is solved by mapping between input and output data using AutoML.  

  1. I would suggest to remove the sentence "Incomplete or outdated information: The study may include something other than the most recent or relevant information, data, or literature related to the research question of blockchain technology." Why this may happen? Why only to blockchain technology? 

ANS – 5. Thank you for your carefully reading. The manuscript was rearranged based on your suggestions.  

  1. Could you please add some references for GDPR and IIoT. 

ANS – 6. Thank you for your remarks. 3 references for GDPR and IIoT were added based on your suggestions.  Specifically,

  1. Golightly, K. Wnuk, N. Shanmugan, A. Shaban, J. Longstaff and V. Chang, "Towards a Working Conceptual Framework: Cyber Law for Data Privacy and Information Security Management for the Industrial Internet of Things Application Domain," 2022 International Conference on Industrial IoT, Big Data and Supply Chain (IIoTBDSC), Beijing, China, 2022, pp. 86-94, doi: 10.1109/IIoTBDSC57192.2022.00027.

  1. Barati, O. Rana, I. Petri and G. Theodorakopoulos, "GDPR Compliance Verification in Internet of Things," in IEEE Access, vol. 8, pp. 119697-119709, 2020, doi: 10.1109/ACCESS.2020.3005509.

  1. K. Kaneen and E. G. M. Petrakis, “Towards evaluating GDPR compliance in IoT applications,” Procedia Comput. Sci., vol. 176, pp. 2989–2998, Jan. 2020, doi: 10.1016/j.procs.2020.09.204.

  1. Figure 1 has a typo (bossines)

ANS – 7. Thank you for your carefully reading. I can’t find the typo (bossiness). I am sorry.   

  1. Page 398 (repetition of word "the")

ANS – 8. Thank you for your carefully reading. The manuscript was rearranged based on your suggestions.  

Reviewer 4 Report (New Reviewer)

Comments to Author

The authors provided an overview of contemporary approaches and solutions for addressing privacy risks in the IIoT, such as encryption and anonymization techniques, access controls and authorization mechanisms, and monitoring and surveillance technologies. My comment to the authors as follows:

1) The introduction should be extended to highlight the contribution in a clear list.

2) The authors need to separate one section (section 2) with the title "related works" in which they should explain all related works and advantages of their review work.

3) I recommend authors to add number 5 of Dimensions of Privacy which is non-repudiation at the end of section 3. Of course, you need to explain how the non-repudiation affects the privacy.

 Authors need to paraphrase some wrong meaning/structure sentences as follows:

1) 

 "However, the implementation of IIoT systems is not without its challenges, particularly in terms of security and privacy".

2) 

" In a world where humans and machines must collaborate and coexist, traditional  business organizational structures create barriers and obstacles that not only waste energy but also devalue information and impede knowledge diffusion.".

3) 

 " recording, use, storage, processing, sharing or sharing"

Author Response

The authors provided an overview of contemporary approaches and solutions for addressing privacy risks in the IIoT, such as encryption and anonymization techniques, access controls and authorization mechanisms, and monitoring and surveillance technologies. My comment to the authors as follows:

1) The introduction should be extended to highlight the contribution in a clear list.

Thank you for this comment. We added the following explanations: “In summary, this study makes the following significant contributions to understanding and implementing IIoT in the industrial context:

  1. IIoT's Transformative Impact on Industrial Organizations: The introduction emphasizes how the Industrial Internet of Things (IIoT) is revolutionizing the manufacturing industry by facilitating ubiquitous connectivity and autonomous data exchange. This transformation is breaking down barriers between physical and digital realms, leading to remarkable progress in industrial operations and modern business practices.
  2. Integration of AI and Industry 4.0 in IIoT: The paper highlights that IIoT represents a forward-thinking industrial technology that combines two digitization strategies - Artificial Intelligence (AI) and Industry 4.0. This integration of data analysis and high-level automation significantly enhances the industrial environment and operations.
  3. Challenges in IIoT Implementation: The introduction acknowledges the challenges associated with IIoT, particularly in terms of security. The complex and heterogeneous nature of IIoT systems makes them vulnerable to sophisticated security attacks at various levels of networking and communication architecture. These challenges can lead to mistrust in network operations, privacy breaches, and the loss of critical data.
  4. Advantages and Disadvantages of IIoT Devices: The introduction presents a clear and concise list outlining the advantages and disadvantages of IIoT devices in Table 1. The benefits include enhanced efficiency, improved data collection, cost savings, and remote monitoring. On the other hand, the disadvantages include security risks, compatibility issues, lack of standardization, and high implementation costs.
  5. Focus on Industrial Privacy in IIoT Ecosystems: The research aims to address the crucial aspect of industrial privacy in an IIoT ecosystem. It defines industrial privacy as the protection of sensitive information and personal data within industrial settings, such as manufacturing plants and supply chains. The research emphasizes the need to implement security measures, policies, and practices to safeguard confidential information from unauthorized access and misuse.
  6. Identification of Privacy Dimensions: The study delves into the identification and definition of privacy dimensions within the IIoT ecosystem. These dimensions represent different aspects or components of privacy that are considered when addressing privacy concerns. By understanding the multifaceted nature of privacy, the research aims to guide the development of privacy frameworks and practices.
  7. Techniques and Best Practices for Data Privacy and Security: The paper explores contemporary techniques, technologies, and best practices used to ensure data privacy and security in IIoT systems. It includes an analysis of the latest methodologies and tools implemented to maintain data confidentiality, integrity, and availability in industrial environments.
  8. Establishing a Safe and Private IIoT Ecosystem: The research aims to identify how organizations can establish an ideal, safe, and private IIoT ecosystem within the industrial domain. It delves into various considerations and factors that need to be taken into account to create a secure environment. Additionally, the study offers recommendations on implementing privacy-enhancing measures and adhering to industry standards to effectively manage privacy risks and regulatory compliance.”

2) The authors need to separate one section (section 2) with the title "related works" in which they should explain all related works and advantages of their review work.

Thank you for this comment. The revised paper includes the new section “Related Works”: The complexity and heterogeneity of IIoT systems make ensuring availability, confidentiality, and integrity difficult. As a result, there is a risk of mistrust in network operations, privacy concerns, and the potential loss of critical personal and sensitive information of network end-users. The above challenges have caused widespread concern in the scientific community, prompting several solutions to be proposed at various levels. For example, existing methods for protecting location privacy are mostly based on traditional anonymization, fuzzy, and cryptography technology, with little success in the big data environment, for example, sensor networks contain sensitive information that must be appropriately protected. Current trends, such as "Industrie 4.0" and the IIoT, generate, process, and exchange massive amounts of security-critical and privacy-sensitive data, making them appealing targets for cyber-attacks. However, previous methods ignored the issue of privacy protection, resulting in a violation of privacy. In this paper [5], the authors propose a location privacy protection method that meets the differential privacy constraint while also maximizing the utility of data and algorithms in IIoT. Because location data has a high value but a low density, the authors combine utility and privacy to create a multilevel location information tree model. Furthermore, the differential privacy index mechanism is used to select data based on the frequency with which tree nodes access data. Finally, the Laplace scheme is used to introduce noise into the data accessing frequency. As demonstrated by the theoretical analysis and experimental results, the proposed strategy can significantly improve security, privacy, and applicability.

A tensor-based multiple clustering method has recently been created with the goal of uncovering hidden distinct data structures in large data from diverse perspectives, and it may be widely employed in IIoT to improve production and service quality. Yet, due to the high computational cost and massive volume of data, outsourcing processing to relatively low-cost cloud servers can significantly reduce local costs, but there is a considerable risk of revealing user privacy. To overcome the aforementioned issue, a secure hybrid cloud-based tensor-based multiple clustering method is proposed [6]. The proposed technique encrypts object tensors using a homomorphic cryptosystem and then uses cloud servers to completely conduct various clustering calculations over encrypted object tensors. A set of related security subprotocols is also developed to facilitate privacy-preserving tensor-based multiple clusterings. Just encryption and perturbation removal are performed on the client in the proposed method, making it very lightweight for consumers. Experiment findings show that the suggested approach is accurate and efficient when grouping items into different groups, with no leakage of private or supplementary information. Furthermore, when more cloud nodes are used, the technique offers excellent scalability, making it ideal for clustering Industrial IoT big data.

In addition, clients that are equipped with growing cloud computing choose to outsource an increasing amount of IIoT data to the cloud to alleviate the high storage and processing strain. Existing searchable encryption (SE) systems, however, only apply to IIoT records with textual keyword fields, rather than those with both digital and textual keyword fields. Furthermore, due to the significant key storage expense, the key management issue continues to limit the practicality and availability of SE schemes. To that purpose, the authors [7] describe an outsourced Hybrid Keyword-Field Search over Encrypted Data with Efficient Key Management (HKFS-KM) system that makes use of the relevance score function and a keyed hash tree. Formal security study demonstrates that the HKFS-KM scheme achieves keyword privacy and trapdoor unlinkability in both known ciphertexts and known background attack models. The experimental findings utilizing real-world datasets demonstrate its efficiency and applicability in practice.

Although cyber-attacks on the Industrial Internet of Things (IIoT) continue to be a serious concern, blockchain has emerged as a viable technology for IIoT security due to its decentralization and immutability. Current blockchain designs, on the other hand, provide considerable computational complexity and latency concerns, making them unsuitable for IIoT. Xyreum, a novel high-performance and scalable blockchain for increased IIoT security and privacy, is proposed in this research [8]. To accomplish Mutual Multi-Factor Authentication, Xyreum employs a Time-based Zero-Knowledge Proof of Knowledge (T-ZKPK) with authenticated encryption (MMFA). T-ZKPK characteristics are also utilized to help with Key Establishment for transaction security. Their approach to establishing consensus, which is a group decision-making process on the blockchain, is based on lightweight cryptographic techniques. They test their scheme for security, privacy, and performance, and the results show that, when compared to existing relevant blockchain solutions, their scheme is secure, privacy-preserving, and achieves a significant reduction in computation complexity and latency performance while maintaining high scalability. They also demonstrate a blockchain-based security protocol used in a variety of application domains.

Moreover, deep learning offers a great chance to extract usable knowledge from massive amounts of data in IIoT. Yet, the absence of large public datasets will result in poor performance and overfitting of the learnt model. As a result, federated deep learning across distant datasets has been proposed. Yet, it invariably brings new security challenges, such as revealing participant data privacy. Existing solutions, however, cannot ensure the privacy of each participant's data in a learning group. The authors [9] suggest two privacy-preserving asynchronous deep learning systems in this article: DeepPAR (privacy-preserving and asynchronous deep learning via re-encryption) and DeepDPA (dynamic privacy-preserving and asynchronous deep learning). In comparison to previous work, DeepPAR secures each participant's input privacy while maintaining dynamic update secrecy naturally. Meanwhile, DeepDPA allows for the backward privacy of group participants to be guaranteed in a lightweight manner. Security analyses and performance tests on real-world datasets demonstrate that their suggested systems are safe, efficient, and effective.

On the other hand, the rapid increase in the volume of data created by connected devices in the IIoT paradigm brings up new opportunities for improving service quality for developing applications through data sharing. Yet, data providers have significant challenges in sharing their data through wireless networks due to security and privacy concerns (e.g., data leakage). Private data leaks can cause major problems beyond financial loss for companies. The authors [10] first design a blockchain-powered safe data-sharing architecture for distributed multiple parties in this study. Finally, using privacy-protected federated learning, they transform the data-sharing challenge into a machine-learning problem. Data privacy is protected by providing the data model rather than releasing the actual data. Lastly, they incorporate federated learning into the permissioned blockchain consensus process, such that the computing work for consensus can also be used for federated training. The suggested data-sharing strategy achieves good accuracy, high efficiency, and better security, according to numerical results generated from real-world datasets.

Finally, the advantage of using edge computing is that it may be used as a complement to cloud computing; blockchain is an alternative for creating a transparent safe environment for data storage/governance. Instead, than employing these two strategies separately, the authors propose a novel methodology in this article [11] termed the blockchain-based Internet of Edge model, which merges IIoT with edge computing and blockchain. The suggested approach, intended for a scalable and controllable IIoT system, takes advantage of the benefits of edge computing and blockchain to construct a privacy-preserving mechanism while taking into account other constraints such as energy cost. They carry out experiment evaluations on Ethereum. According to their data gathering, the proposed strategy improves privacy safeguards while reducing energy consumption.

In the rapidly evolving landscape of the IIoT, the research community as mentioned above has embraced various privacy methods to protect sensitive data and ensure secure operations. However, each method comes with its own set of disadvantages, highlighting the necessity for a holistic approach when addressing privacy dimensions in the IIoT ecosystem. From this point of view, the IIoT ecosystem's privacy dimensions require a comprehensive approach that takes into account the interconnectedness of various methods and their associated advantages. By presenting a holistic approach to privacy in the IIoT, industries and organizations can strengthen their defenses against evolving threats and protect sensitive data, ensuring the secure and sustainable growth of IIoT technologies.

3) I recommend authors to add number 5 of Dimensions of Privacy which is non-repudiation at the end of section 3. Of course, you need to explain how the non-repudiation affects the privacy.

Thank you for this comment. The revised paper includes the following:

  1. Non-repudiation: It is a security property that ensures the sender of a message, or a digital transaction cannot later deny sending the message or engaging in the transaction. It provides proof that a particular action or communication occurred and prevents individuals from disowning their actions or denying their involvement. In the context of privacy, non-repudiation plays a significant role in maintaining trust and accountability in digital interactions. Specifically:
    1. Digital Transactions: It ensures that both parties involved in a transaction cannot later deny their participation, protecting the privacy of sensitive information exchanged during the transaction.
    2. Message Authenticity: Non-repudiation guarantees that the sender of a message cannot deny sending it. This property is particularly essential when exchanging private or confidential messages. It helps prevent unauthorized access and ensures that the recipient can trust the origin and authenticity of the message.
    3. Digital Signatures: Digital signatures are a cryptographic mechanism used for non-repudiation. By using digital signatures, individuals can sign electronic documents or messages, providing assurance that the content remains unchanged, and that the signer cannot later deny their approval.
    4. Legal Implications: Non-repudiation can have legal implications in contracts and agreements. If a digital transaction or communication has non-repudiation measures in place, it can serve as evidence in case of disputes, protecting the privacy of individuals involved by establishing their roles in the interaction.

Comments on the Quality of English Language

Authors need to paraphrase some wrong meaning/structure sentences as follows:

1)  "However, the implementation of IIoT systems is not without its challenges, particularly in terms of security and privacy".

2) " In a world where humans and machines must collaborate and coexist, traditional  business organizational structures create barriers and obstacles that not only waste energy but also devalue information and impede knowledge diffusion.".

3)  " recording, use, storage, processing, sharing or sharing"

Your attention is greatly appreciated. The aforementioned issues have been rectified.

Round 2

Reviewer 1 Report (New Reviewer)

Accept in current format

Author Response

We deeply appreciate your time and effort in reviewing our manuscript. 

Reviewer 2 Report (New Reviewer)

A number of improvements was done to this manuscript by mostly shuffling the content between the sections (in comparison to the previous version of the paper).
Also, numerous previously provided comments were either not addressed appropriately or were skipped all together.
Authors, once again, are advised to rethink what exactly they intend to demonstrate within this manuscript and think about a continuous story with decent explanation of newly appearing terms and their relevance to the discussion. It is highly advised to shorten up the manuscript and remove all repetitive elements, define the research questions that are guiding this work, include comparison with other review works in the same domain, re-structure their findings. In the current state this manuscript still has a very doubtful scientific value and needs significant improvements prior to the publication.

Below is a recap:

1. What are the research questions?
2. What are the scientific contributions of the authors, how their investigation is supported, what was the methodology?
3. The language needs major improvement from grammatical point of view and also how the sentences are composed.
4. It is very difficult to follow  the logic and story line of the manuscript.
5. No related works is present in the manuscript.
6. Many statements are either not reasoned or no references are provided or are simply misleading. The content is overlapping between sections and within sections. Some examples:
- lines 41-43 - no reasoning
- lines 61-63 - it is not clear in table 1 which columns/content refer to benefits and what to challenges. Instead it lists advantages and disadvantages.
- line 70 - what is "natural people"?
- lines 66-68 - repetition of previous paragraph
- line 73 - what is "industrial Privacy" - this term was not previously introduced.
- lines 76-83 - repetitions of previous paragraphs
- line 91 - what is "privacy dimension" - this term was not previously introduced.
- lines 109-114 - repetition of the contributions summary
- lines 116-121 - repetitions of text from previous section
- lines 137-140 - what is the relevance of UAVs here? This comment was not addressed from the first review.
- lines 141-185 - introduction of AI is not clear. What is the relevance to discuss AI in this section (IIoT and privacy preserving architectures)? This comment was not addressed from the first review.
- lines 186-268 - what is the relevance of IIoT trends to this section?
- lines 197-198 - previous comment on "local sockets" still stands. Even if definition was provided, the use of the term in the current context still does not make sense.
- line 276 and figure 1 - the relevance of this figure is not clear. Comment still not addressed from the previous review. Furthermore, consecutive paragraph with discussion on ADVOCATE appears out of blue and has no logical link with previously discussed notions.
- lines 295-346 - no references to support the statements. If all architectures come from [12], it is inappropriate way to cite previous work by extracting large chunk of content of existing work
- lines 354-381 - same as in previous comment.
- lines 382-387 - repetition of the content
- lines 394-426 - no referencing provided
- lines 520-582 - current structure of the presented analysis is not clear. Possible prepare a table instead to synthesise more clearly.
- line 597 - what is ".U.E.U."?
- lines 644-647 - repetition
- lines 648-681 - no referencing provided
- lines 693-706 - how these privacy dimensions are different from those introduced at the beginning of the section. The link and reasoning is not clear. This is a comment from the previous review, still not addressed.
- line 710 - what is the purpose and scope of this section? How is it different from previous sections. It looks rather as a repetition of previous discussions. The structure of this section is not clear.
- line 801 - what is "IP-based equipment" here? The "IP" term was not introduced.
- lines 804-808 - repetition
- lines 815-842 - no referencing provided
- lines 866-869 and figure 4 - has no relevance to the section's intended scope, which is "privacy threats"
- lines 875-885 - it is not clear if this paragraph still refers to BANTA.
- lines 886-925 - this part rather refers to recommendations, not to the "privacy threats". The linking and logical story line is not clear.
- lines 948-962 - no referencing provided.
- section 6 - the structure is confusing. First list is "requirements - recommendations". By definition, those are two different notions. Decide which one you outline. Furthermore, section gives another list of recommendation is given, followed by another list of "requirements and suggestions for IoT implementations". This structure is very confusing and does not show a though through unified approach to present clearly the requirements and recommendations.
- lines 1227-1233 - authors claim that their work's emphasis is on "protecting personal data in the context of criminal investigations or the execution of criminal sanctions". Yet the only place where criminal investigation were mentioned are introduction section and section 3, line 625.
- line 1239 - authors claim that "have differentiated from other review papers", yet there is no review of other similar works in this paper.
- lines 1295-1305 - the relevance of blockchain here is not clear.

Please comments above

Author Response

Dear respected Reviewer,

We deeply appreciate your time and effort in reviewing our manuscript. Your comments are very helpful for revising and improving our paper. We have revised the manuscript considering all the insightful comments to enhance the paper's readability. We believe these changes have strengthened the rationale and importance of our study.

A number of improvements was done to this manuscript by mostly shuffling the content between the sections (in comparison to the previous version of the paper).
Also, numerous previously provided comments were either not addressed appropriately or were skipped all together.
Authors, once again, are advised to rethink what exactly they intend to demonstrate within this manuscript and think about a continuous story with decent explanation of newly appearing terms and their relevance to the discussion. It is highly advised to shorten up the manuscript and remove all repetitive elements, define the research questions that are guiding this work, include comparison with other review works in the same domain, re-structure their findings. In the current state this manuscript still has a very doubtful scientific value and needs significant improvements prior to the publication.

Below is a recap:

1. What are the research questions?

Thank you for this comment. The revised introduction includes the appropriate details based on your suggestions.  

  1. What are the scientific contributions of the authors, how their investigation is supported, what was the methodology?

Thank you for this comment. The revised introduction includes the appropriate details based on your suggestions.  

  1. The language needs major improvement from grammatical point of view and also how the sentences are composed.

Thank you for your remarks. Typos and grammar errors have been corrected, and the English language used throughout the manuscript has been improved.

  1. It is very difficult to follow  the logic and story line of the manuscript.

Thank you for this constructive comment. Based on the reviewer's suggestions, we reorganized the entire paper. The paper now reads much better, and the work presented has been enhanced to a level appropriate for this journal's readership and scientific standing.

  1. No related works is present in the manuscript.

Thank you for your suggestions. Because the research study is highly specific and investigates a niche area, the number of relevant prior studies or publications is limited.

  1. Many statements are either not reasoned or no references are provided or are simply misleading. The content is overlapping between sections and within sections. Some examples:
    - lines 41-43 - no reasoning

Thank you for your feedback. The text was removed.

- lines 61-63 - it is not clear in table 1 which columns/content refer to benefits and what to challenges. Instead it lists advantages and disadvantages.

Thank you for your comment. Table 1 summarizes the advantages and disadvantages of IIoT devices as requested by examiners and editors during a preview review round.  The text about the benefits and challenges has been removed.

- line 70 - what is "natural people"?

Thank you for your feedback. The “natural” was removed.

- lines 66-68 - repetition of previous paragraph

Thank you for your feedback. The text was removed.

- line 73 - what is "industrial Privacy" - this term was not previously introduced.

Thank you for this comment. The appropriate details were added.  

- lines 76-83 - repetitions of previous paragraphs

Thank you for your feedback. The text was removed.

- line 91 - what is "privacy dimension" - this term was not previously introduced.

Thank you for this comment. The appropriate details were added.  

- lines 109-114 - repetition of the contributions summary

Thank you for your remarks. The text was removed.

- lines 116-121 - repetitions of text from previous section

Thank you for your remarks. The text was removed.

- lines 137-140 - what is the relevance of UAVs here? This comment was not addressed from the first review.

Thank you for your remarks. The text was removed.

- lines 141-185 - introduction of AI is not clear. What is the relevance to discuss AI in this section (IIoT and privacy preserving architectures)? This comment was not addressed from the first review.

Thank you for your remarks. The text was removed.

- lines 186-268 - what is the relevance of IIoT trends to this section?

Thank you for this constructive comment. The appropriate details were added.  

- lines 197-198 - previous comment on "local sockets" still stands. Even if definition was provided, the use of the term in the current context still does not make sense.

Thank you for this constructive comment. The appropriate details were added.  

- line 276 and figure 1 - the relevance of this figure is not clear. Comment still not addressed from the previous review. Furthermore, consecutive paragraph with discussion on ADVOCATE appears out of blue and has no logical link with previously discussed notions.

Thank you for this constructive comment. The appropriate details were added.  

- lines 295-346 - no references to support the statements. If all architectures come from [12], it is inappropriate way to cite previous work by extracting large chunk of content of existing work

Thank you for your careful reading. The citations have been added.

- lines 354-381 - same as in previous comment.

Thank you for your careful reading. The citations have been added.

- lines 382-387 - repetition of the content

Thank you for your remarks. The text was removed.

- lines 394-426 - no referencing provided

Thank you for your careful reading. The citations have been added.

- lines 520-582 - current structure of the presented analysis is not clear. Possible prepare a table instead to synthesise more clearly.

Thank you for this constructive comment. The table was constructed based on your suggestions..  

- line 597 - what is ".U.E.U."?

Thank you for your careful reading. The abbreviation was revised as EU.

- lines 644-647 – repetition

Thank you for this constructive comment. The text was revised.  

- lines 648-681 - no referencing provided

Thank you for your careful reading. The citations have been added.

- lines 693-706 - how these privacy dimensions are different from those introduced at the beginning of the section. The link and reasoning is not clear. This is a comment from the previous review, still not addressed.

Thank you for your suggestions. These dimensions provide a framework for understanding the privacy needs of IIoT ecosystems and guide the development of privacy settings and strategies. The text was revised based on your remarks.  

- line 710 - what is the purpose and scope of this section? How is it different from previous sections. It looks rather as a repetition of previous discussions. The structure of this section is not clear.

Thank you for this helpful comment. Industrial privacy refers to the protection of sensitive information and trade secrets within the context of industrial and manufacturing sectors. It involves safeguarding confidential data, proprietary processes, intellectual property, and other critical business information from unauthorized access, theft, or misuse by competitors, employees, or external entities. The text was revised based on your suggestions.  

- line 801 - what is "IP-based equipment" here? The "IP" term was not introduced.

Thank you for this constructive comment. The text was revised.  

- lines 804-808 – repetition

Thank you for this constructive comment. The text was revised.  

- lines 815-842 - no referencing provided

Thank you for your careful reading. The citation has been added.

- lines 866-869 and figure 4 - has no relevance to the section's intended scope, which is "privacy threats"

Thank you for this comment. Figure 4 depicts a blockchain-based system that combines a multi-signature system and differential privacy algorithms to provide a secure and privacy-preserving solution for network traffic analysis called BANTA.

- lines 875-885 - it is not clear if this paragraph still refers to BANTA.

Thank you for this constructive comment. The text was revised.  

- lines 886-925 - this part rather refers to recommendations, not to the "privacy threats". The linking and logical story line is not clear.

Thank you for your remarks. The text was removed.

- lines 948-962 - no referencing provided.

Thank you for taking the time to read this. The citation has been added.

- section 6 - the structure is confusing. First list is "requirements - recommendations". By definition, those are two different notions. Decide which one you outline. Furthermore, section gives another list of recommendation is given, followed by another list of "requirements and suggestions for IoT implementations". This structure is very confusing and does not show a though through unified approach to present clearly the requirements and recommendations.

Thank you for your suggestions. In the revised manuscript this section was split into Section 6 Privacy Requirements and Section 7 Recommendations. 

- lines 1227-1233 - authors claim that their work's emphasis is on "protecting personal data in the context of criminal investigations or the execution of criminal sanctions". Yet the only place where criminal investigation were mentioned are introduction section and section 3, line 625.

Thank you for your feedback. The text was removed.

- line 1239 - authors claim that "have differentiated from other review papers", yet there is no review of other similar works in this paper.

Thank you for this constructive comment. In the revised manuscript the text was removed.  

- lines 1295-1305 - the relevance of blockchain here is not clear.

Thank you for this comment. The text was removed.

Reviewer 3 Report (New Reviewer)

The authors have addressed the received comments. Although i would suggest to the authors to be careful and always find time to check the comments received. In my typo comment (for the erroneous "bossines") I was mistakenly referring to figure 1 while the error was in figure 2. However the authors were not able to discover this! 

No comments

Author Response

We deeply appreciate your time and effort in reviewing our manuscript. We are sorry for your previous comment is not faced correctly.  We have revised the manuscript considering your insightful suggestion. Thank you for your careful reading. 

Round 3

Reviewer 2 Report (New Reviewer)

Unfortunately, previously provided feedback, once again, was not addressed appropriately. 
In particular, the following comments still stand:

1. What are the research questions?
2. What are the scientific contributions of the authors, how their investigation is supported, what was the methodology?
3. The language needs major improvement from grammatical point of view and also how the sentences are composed.
4. It is very difficult to follow  the logic and story line of the manuscript.
5. No related works is present in the manuscript.

The answers to these questions should be provided in a clear way within the manuscript, implying significant changes to the outline of the manuscript.

Authors, once again, are advised to rethink what exactly they intend to demonstrate within this manuscript and think about a continuous story with decent explanation of newly appearing terms and their relevance to the discussion. It is highly advised to shorten up the manuscript and remove all repetitive elements, define the research questions that are guiding this work, include comparison with other review works in the same domain, re-structure their findings. In the current state this manuscript still has a very doubtful scientific value and needs significant improvements prior to the publication.

-

Author Response

Unfortunately, previously provided feedback, once again, was not addressed appropriately. 
In particular, the following comments still stand:

  1. What are the research questions?

Thank you for this comment. The revised introduction includes the appropriate details based on your suggestions.  We added the following explanations:” This research endeavors to address several key aspects concerning the privacy requirements within an IIoT ecosystem, particularly in relation to the processing of personal data by competent authorities. Specifically, the research questions addressed in this study are:

  1. What are the specific industrial privacy requirements within an IIoT ecosystem, particularly in relation to the processing of personal data by competent authorities?
  2. What are the existing approaches and solutions used to mitigate privacy risks in industrial settings?
  3. How can privacy dimensions be identified and defined to safeguard individuals within the IIoT ecosystem? What are the different aspects or components of privacy that should be considered?
  4. What are the contemporary techniques, technologies, and best practices employed to ensure data privacy and security in IIoT systems within industrial environments?
  5. How can organizations establish an ideal, safe, and private IIoT ecosystem within the industrial domain?
  6. What considerations and factors need to be taken into account to create a secure environment within the industrial domain while adhering to relevant industry standards?
  7. What are the recommendations for implementing privacy-enhancing measures in IIoT systems to effectively manage privacy risks and ensure compliance with relevant regulations?

By addressing these research questions, the study aims to provide insights into the challenges and strategies for maintaining industrial privacy within the context of IIoT ecosystems. It also seeks to offer guidance to organizations in enhancing data protection measures, thereby fostering trust, compliance, and sustainable growth within the industrial sectors.

  1. What are the scientific contributions of the authors, how their investigation is supported, what was the methodology?

Thank you for this comment. We added the following explanations: “This study aims to investigate the privacy requirements of an IIoT ecosystem as outlined by industry standards. It provides a comprehensive overview of the IIoT, its advantages, disadvantages, challenges, and the imperative need for industrial privacy. The research methodology encompasses a thorough literature review to gather existing knowledge and insights on the subject. Additionally, it explores how IIoT is transforming the manufacturing industry and enhancing industrial processes, incorporating case studies and real-world examples to illustrate its practical applications and impact. Also, the research endeavors to offer actionable recommendations on implementing privacy-enhancing measures and establishing a secure IIoT ecosystem.”

In summary, this study makes the following significant contributions to understanding and implementing IIoT in the industrial context:

  1. IIoT's Transformative Impact on Industrial Organizations: The introduction emphasizes how the Industrial Internet of Things (IIoT) is revolutionizing the manufacturing industry by facilitating ubiquitous connectivity and autonomous data exchange. This transformation is breaking down barriers between physical and digital realms, leading to remarkable progress in industrial operations and modern business practices.
  2. Integration of AI and Industry 4.0 in IIoT: The paper highlights that IIoT represents a forward-thinking industrial technology that combines two digitization strategies - Artificial Intelligence (AI) and Industry 4.0. This integration of data analysis and high-level automation significantly enhances the industrial environment and operations.
  3. Challenges in IIoT Implementation: The introduction acknowledges the challenges associated with IIoT, particularly in terms of security. The complex and heterogeneous nature of IIoT systems makes them vulnerable to sophisticated security attacks at various levels of networking and communication architecture. These challenges can lead to mistrust in network operations, privacy breaches, and the loss of critical data.
  4. Advantages and Disadvantages of IIoT Devices: The introduction presents a clear and concise list outlining the advantages and disadvantages of IIoT devices in Table 1. The benefits include enhanced efficiency, improved data collection, cost savings, and remote monitoring. On the other hand, the disadvantages include security risks, compatibility issues, lack of standardization, and high implementation costs.
  5. Focus on Industrial Privacy in IIoT Ecosystems: The research aims to address the crucial aspect of industrial privacy in an IIoT ecosystem. It defines industrial privacy as the protection of sensitive information and personal data within industrial settings, such as manufacturing plants and supply chains. The research emphasizes the need to implement security measures, policies, and practices to safeguard confidential information from unauthorized access and misuse.
  6. Identification of Privacy Dimensions: The study delves into the identification and definition of privacy dimensions within the IIoT ecosystem. These dimensions represent different aspects or components of privacy that are considered when addressing privacy concerns. By understanding the multifaceted nature of privacy, the research aims to guide the development of privacy frameworks and practices.
  7. Techniques and Best Practices for Data Privacy and Security: The paper explores contemporary techniques, technologies, and best practices used to ensure data privacy and security in IIoT systems. It includes an analysis of the latest methodologies and tools implemented to maintain data confidentiality, integrity, and availability in industrial environments.
  8. Establishing a Safe and Private IIoT Ecosystem: The research aims to identify how organizations can establish an ideal, safe, and private IIoT ecosystem within the industrial domain. It delves into various considerations and factors that need to be taken into account to create a secure environment. Additionally, the study offers recommendations on implementing privacy-enhancing measures and adhering to industry standards to effectively manage privacy risks and regulatory compliance.”

  9. The language needs major improvement from grammatical point of view and also how the sentences are composed.

Your careful attention is greatly appreciated. We've addressed the reviewer's suggestions, making adjustments to the entire paper. Typos and grammar mistakes have been rectified, and the manuscript's English language has been upgraded. The paper's readability has significantly improved, and the presented work has been elevated to meet the standards of this journal's audience and scientific community.

  1. It is very difficult to follow  the logic and story line of the manuscript.

Thank you for this comment. We reorganized the entire paper based on the reviewer's suggestions.

  1. No related works is present in the manuscript.

Thank you for this comment. The revised paper includes the new section “Related Works”: The complexity and heterogeneity of IIoT systems make ensuring availability, confidentiality, and integrity difficult. As a result, there is a risk of mistrust in network operations, privacy concerns, and the potential loss of critical personal and sensitive information of network end-users. The above challenges have caused widespread concern in the scientific community, prompting several solutions to be proposed at various levels. For example, existing methods for protecting location privacy are mostly based on traditional anonymization, fuzzy, and cryptography technology, with little success in the big data environment, for example, sensor networks contain sensitive information that must be appropriately protected. Current trends, such as "Industrie 4.0" and the IIoT, generate, process, and exchange massive amounts of security-critical and privacy-sensitive data, making them appealing targets for cyber-attacks. However, previous methods ignored the issue of privacy protection, resulting in a violation of privacy. In this paper [5], the authors propose a location privacy protection method that meets the differential privacy constraint while also maximizing the utility of data and algorithms in IIoT. Because location data has a high value but a low density, the authors combine utility and privacy to create a multilevel location information tree model. Furthermore, the differential privacy index mechanism is used to select data based on the frequency with which tree nodes access data. Finally, the Laplace scheme is used to introduce noise into the data accessing frequency. As demonstrated by the theoretical analysis and experimental results, the proposed strategy can significantly improve security, privacy, and applicability.

A tensor-based multiple clustering method has recently been created with the goal of uncovering hidden distinct data structures in large data from diverse perspectives, and it may be widely employed in IIoT to improve production and service quality. Yet, due to the high computational cost and massive volume of data, outsourcing processing to relatively low-cost cloud servers can significantly reduce local costs, but there is a considerable risk of revealing user privacy. To overcome the aforementioned issue, a secure hybrid cloud-based tensor-based multiple clustering method is proposed [6]. The proposed technique encrypts object tensors using a homomorphic cryptosystem and then uses cloud servers to completely conduct various clustering calculations over encrypted object tensors. A set of related security subprotocols is also developed to facilitate privacy-preserving tensor-based multiple clusterings. Just encryption and perturbation removal are performed on the client in the proposed method, making it very lightweight for consumers. Experiment findings show that the suggested approach is accurate and efficient when grouping items into different groups, with no leakage of private or supplementary information. Furthermore, when more cloud nodes are used, the technique offers excellent scalability, making it ideal for clustering Industrial IoT big data.

In addition, clients that are equipped with growing cloud computing choose to outsource an increasing amount of IIoT data to the cloud to alleviate the high storage and processing strain. Existing searchable encryption (SE) systems, however, only apply to IIoT records with textual keyword fields, rather than those with both digital and textual keyword fields. Furthermore, due to the significant key storage expense, the key management issue continues to limit the practicality and availability of SE schemes. To that purpose, the authors [7] describe an outsourced Hybrid Keyword-Field Search over Encrypted Data with Efficient Key Management (HKFS-KM) system that makes use of the relevance score function and a keyed hash tree. Formal security study demonstrates that the HKFS-KM scheme achieves keyword privacy and trapdoor unlinkability in both known ciphertexts and known background attack models. The experimental findings utilizing real-world datasets demonstrate its efficiency and applicability in practice.

Although cyber-attacks on the Industrial Internet of Things (IIoT) continue to be a serious concern, blockchain has emerged as a viable technology for IIoT security due to its decentralization and immutability. Current blockchain designs, on the other hand, provide considerable computational complexity and latency concerns, making them unsuitable for IIoT. Xyreum, a novel high-performance and scalable blockchain for increased IIoT security and privacy, is proposed in this research [8]. To accomplish Mutual Multi-Factor Authentication, Xyreum employs a Time-based Zero-Knowledge Proof of Knowledge (T-ZKPK) with authenticated encryption (MMFA). T-ZKPK characteristics are also utilized to help with Key Establishment for transaction security. Their approach to establishing consensus, which is a group decision-making process on the blockchain, is based on lightweight cryptographic techniques. They test their scheme for security, privacy, and performance, and the results show that, when compared to existing relevant blockchain solutions, their scheme is secure, privacy-preserving, and achieves a significant reduction in computation complexity and latency performance while maintaining high scalability. They also demonstrate a blockchain-based security protocol used in a variety of application domains.

Moreover, deep learning offers a great chance to extract usable knowledge from massive amounts of data in IIoT. Yet, the absence of large public datasets will result in poor performance and overfitting of the learnt model. As a result, federated deep learning across distant datasets has been proposed. Yet, it invariably brings new security challenges, such as revealing participant data privacy. Existing solutions, however, cannot ensure the privacy of each participant's data in a learning group. The authors [9] suggest two privacy-preserving asynchronous deep learning systems in this article: DeepPAR (privacy-preserving and asynchronous deep learning via re-encryption) and DeepDPA (dynamic privacy-preserving and asynchronous deep learning). In comparison to previous work, DeepPAR secures each participant's input privacy while maintaining dynamic update secrecy naturally. Meanwhile, DeepDPA allows for the backward privacy of group participants to be guaranteed in a lightweight manner. Security analyses and performance tests on real-world datasets demonstrate that their suggested systems are safe, efficient, and effective.

On the other hand, the rapid increase in the volume of data created by connected devices in the IIoT paradigm brings up new opportunities for improving service quality for developing applications through data sharing. Yet, data providers have significant challenges in sharing their data through wireless networks due to security and privacy concerns (e.g., data leakage). Private data leaks can cause major problems beyond financial loss for companies. The authors [10] first design a blockchain-powered safe data-sharing architecture for distributed multiple parties in this study. Finally, using privacy-protected federated learning, they transform the data-sharing challenge into a machine-learning problem. Data privacy is protected by providing the data model rather than releasing the actual data. Lastly, they incorporate federated learning into the permissioned blockchain consensus process, such that the computing work for consensus can also be used for federated training. The suggested data-sharing strategy achieves good accuracy, high efficiency, and better security, according to numerical results generated from real-world datasets.

Finally, the advantage of using edge computing is that it may be used as a complement to cloud computing; blockchain is an alternative for creating a transparent safe environment for data storage/governance. Instead, than employing these two strategies separately, the authors propose a novel methodology in this article [11] termed the blockchain-based Internet of Edge model, which merges IIoT with edge computing and blockchain. The suggested approach, intended for a scalable and controllable IIoT system, takes advantage of the benefits of edge computing and blockchain to construct a privacy-preserving mechanism while taking into account other constraints such as energy cost. They carry out experiment evaluations on Ethereum. According to their data gathering, the proposed strategy improves privacy safeguards while reducing energy consumption.

In the rapidly evolving landscape of the IIoT, the research community as mentioned above has embraced various privacy methods to protect sensitive data and ensure secure operations. However, each method comes with its own set of disadvantages, highlighting the necessity for a holistic approach when addressing privacy dimensions in the IIoT ecosystem. From this point of view, the IIoT ecosystem's privacy dimensions require a comprehensive approach that takes into account the interconnectedness of various methods and their associated advantages. By presenting a holistic approach to privacy in the IIoT, industries and organizations can strengthen their defenses against evolving threats and protect sensitive data, ensuring the secure and sustainable growth of IIoT technologies.

This manuscript is a resubmission of an earlier submission. The following is a list of the peer review reports and author responses from that submission.

Round 1

Reviewer 1 Report

In this article, the authors review the dimensions of privacy in the Industrial Internet of Things. This article is written in a rather general way and it is not possible to identify a benefit. Is technically very poorly described. Lacks a discussion section, which would more thoroughly evaluate the sources cited. The paper produced here lacks important details such as its motivation, methodology and logical structure.

1. IIoT devices are widely known to have a security problem. This is nothing new.
2. The article lacks a summary of the advantages/disadvantages of IIoT devices. (comparison tables)
3. What is the contribution of your study ?
4. What are the disadvantages of your study ?
5. Quite a lot of abbreviations are used in the manuscript. To improve readability, it is recommended to use a table to locate all frequently used abbreviations with their descriptions. Some abbreviations are not even explained.
6. Some sentences are too long to follow, so it is recommended to break them into short but meaningful ones to make the manuscript readable.

Author Response

Dear Reviewer

We want to thank the reviewer for their insightful comments and constructive suggestions, which helped us significantly improve the manuscript. For clarity, we have uploaded a copy of the original manuscript with all changes highlighted. Our point-by-point response to the reviewers' comments is attached to this letter. The comments are reproduced, and our answers follow in a different color (red) immediately after. We would also like to express our gratitude for allowing us to resubmit a revised copy of the manuscript.

Reviewer 1

In this article, the authors review the dimensions of privacy in the Industrial Internet of Things. This article is written in a rather general way and it is not possible to identify a benefit. Is technically very poorly described. Lacks a discussion section, which would more thoroughly evaluate the sources cited. The paper produced here lacks important details such as its motivation, methodology and logical structure.

  1. IIoT devices are widely known to have a security problem. This is nothing new.

A1- Thank you for this comment. As mentioned in the revised conclusion section, Industry 4.0 is a concept that involves the digitalization and automation of manufacturing processes, focusing on creating intelligent factories that can adapt to changing demands and optimize production. While the paper does not explicitly discuss Industry 4.0, it focuses on the IIoT, a critical component of Industry 4.0. and privacy dimensions of this.

  1. The article lacks a summary of the advantages/disadvantages of IIoT devices. (comparison tables)

A2-Thank you for this helpful suggestion. A comparison table outlining the advantages and disadvantages of IIoT devices was added in the introduction section in order to be a helpful addition to the paper.

  1. What is the contribution of your study ?

A3- Thank you for your feedback. The motivation and contributions of the paper are highlighted in the revised version of the introduction section.

  1. What are the disadvantages of your study ?

A4- Thank you for your remarks. Some possible limitations or drawbacks of the study are included in the revised version of the conclusion section.

  1. Quite a lot of abbreviations are used in the manuscript. To improve readability, it is recommended to use a table to locate all frequently used abbreviations with their descriptions. Some abbreviations are not even explained.

A-5. Thank you for this constructive comment. All abbreviations used in the text are included at the end of the paper. This will help readers easily find the definitions they need to understand the paper.

  1. Some sentences are too long to follow, so it is recommended to break them into short but meaningful ones to make the manuscript readable.

A6- We reorganized the entire paper based on the reviewer's suggestions. In addition, typos and grammar errors have been corrected, and the English language used throughout the manuscript has been improved. The paper now reads much better, and the work presented has been enhanced to a level appropriate for this journal's readership and scientific standing. Thank you for taking the time to read this carefully.

Reviewer 2 Report

This is a review paper on IIoT Privacy Dimensions.

I think authors need to do more work on this before it get publish. Paper objectives are not mentioned clearly mentioned. What are the findings of authors as a result of this review are not clearly given.

There should be a separate section on literature review that provide the approach presented in literature with their pros and cons. (may be in tabular form)

There should be a section on discussion where authors should discuss the literature review and give their views on it.

Author Response

Dear Reviewer

We want to thank the reviewer for their insightful comments and constructive suggestions, which helped us significantly improve the manuscript. For clarity, we have uploaded a copy of the original manuscript with all changes highlighted. Our point-by-point response to the reviewers' comments is attached to this letter. The comments are reproduced, and our answers follow in a different color (red) immediately after. We would also like to express our gratitude for allowing us to resubmit a revised copy of the manuscript.

Reviewer 2

This is a review paper on IIoT Privacy Dimensions. I think authors need to do more work on this before it get publish.

  1. Paper objectives are not mentioned clearly mentioned.

A1- Thank you for your feedback. The motivation and contributions of the paper are highlighted in the revised version of the introduction section.

  1. What are the findings of authors as a result of this review are not clearly given.

A2-Thank you for your comment. The research study’s findings are highlighted in the revised version of the conclusion section.

  1. There should be a separate section on literature review that provide the approach presented in literature with their pros and cons. (may be in tabular form)

A3-Thank you for your suggestions. Based on your suggestions, the comparison outlining the advantages and disadvantages of IIoT devices is presented in table 1.

  1. There should be a section on discussion where authors should discuss the literature review and give their views on it.

A4-Thank you for your remarks. The revised version of the conclusion section includes the study's discussion.

Reviewer 3 Report

The manuscript provides an overview of Privacy Dimensions in the Industrial IoT. However, several issues need to be addressed in the manuscript.  some of the comments are presented below:  

The contributions are not clear please provide your contributions in bullet points.  

It is preferred to summarize the related works in a taxonomy table which helps the reader to have a comparison between the other work findings.  

It is recommended to have a comparison table between this work and other works and show what the differences are. 

Weirdly, the work lacks any visualizations, please justify  

It is suggested to improve the section about the current research trends and what are the possible future directions to help researchers with research gaps. You can refer to the following work to have clue about the survey formulation doi.org/10.3390/electronics10091043 

Author Response

Dear Reviewer

We want to thank the reviewer for their insightful comments and constructive suggestions, which helped us significantly improve the manuscript. For clarity, we have uploaded a copy of the original manuscript with all changes highlighted. Our point-by-point response to the reviewers' comments is attached to this letter. The comments are reproduced, and our answers follow in a different color (red) immediately after. We would also like to express our gratitude for allowing us to resubmit a revised copy of the manuscript.

Reviewer 3

The manuscript provides an overview of Privacy Dimensions in the Industrial IoT. However, several issues need to be addressed in the manuscript.  some of the comments are presented below: 

  1. The contributions are not clear please provide your contributions in bullet points.

A1- Thank you for your feedback. The motivation and contributions of the paper are highlighted in the revised version of the introduction section.

  1. It is preferred to summarize the related works in a taxonomy table which helps the reader to have a comparison between the other work findings.

A2- Thank you for this insightful comment allowing us to clarify things further. At the end of each section, we summarize the findings from the mentioned studies.

  1. It is recommended to have a comparison table between this work and other works and show what the differences are.

A3- Thank you for your remarks. Industry 4.0 is a concept that involves the digitalization and automation of manufacturing processes, with a focus on creating intelligent factories that can adapt to changing demands and optimize production. While the paper does not explicitly discuss Industry 4.0, it focuses on the Industrial Internet of Things (IIoT), a critical component of Industry 4.0. IIoT is a crucial enabling technology for Industry 4.0, as it provides the necessary infrastructure for connecting devices, sensors, and machines in a factory setting. The applications mentioned in the question are all relevant to Industry 4.0 and demonstrate the diverse range of use cases that IIoT can enable. For example, using UAVs for fire detection in forestry represents a significant advancement in environmental monitoring and management. At the same time, wearables based on shape memory alloys offer new possibilities for monitoring worker health and safety in a factory setting. The concept of digital twins is also relevant to Industry 4.0. It involves creating virtual replicas of physical assets and processes, which can be used for simulation, optimization, and predictive maintenance. The use of multiple autonomous robots in virtual environments to perform tasks in Industry 4.0 represents another exciting application of IIoT, as it allows for the creation of flexible, adaptable manufacturing systems. Overall, the applications mentioned in the question demonstrate the broad scope and potential of IIoT in enabling the digitalization and automation of various industries, including forestry, manufacturing, and healthcare. As such, they represent essential areas for further research and development in Industry 4.0. Detailed explanations were added in the conclusion section.  Also, based on your suggestions, the comparison outlining the advantages and disadvantages of IIoT devices is presented in table 1.

  1. Weirdly, the work lacks any visualizations, please justify

A4- Thank you for your remarks. Visualizations such as tables, figures, and charts can effectively convey complex information clearly and concisely. They can also help to break up large blocks of text and make the paper more visually appealing and engaging. But in this case, visualizations could have been confusing readers in order to identify the key concepts, highlight the relationships between different dimensions of privacy, or provide a summary of the advantages and disadvantages of IIoT devices. Visualizations can help demonstrate the different approaches and solutions for addressing industrial privacy risks but can add bias in the decision-makers. So we prefer the paper not to include visualizations. If the readers need visually oriented can be addressed the appropriate references.

  1. It is suggested to improve the section about the current research trends and what are the possible future directions to help researchers with research gaps. You can refer to the following work to have clue about the survey formulation doi.org/10.3390/electronics10091043

A5-Thank you for your remarks. The revised version of the conclusion section includes the current research trends and what are the possible future directions to help researchers with research gaps.

Reviewer 4 Report

the authors try to investigate the privacy needs of an IoT eco- 22 system as specified by industry standards. 

Motivation is not clear 

Authors should discuss Industry 4.0 with more details and applications such as  Industry 4.0 towards Forestry 4.0: Fire detection use case, Computing in the sky: A survey on intelligent ubiquitous computing for uav-assisted 6g networks and industry 4.0/5.0,  Digital twins collaboration for automatic erratic operational data detection in industry 4.0, Shape memory alloy-based wearables: a review, and conceptual frameworks on HCI and HRI in Industry 4.0,Autonomous Multi-Robot Collaboration in Virtual Environments to Perform Tasks in Industry 4.0.

at the end of every section, authors should summaries the finding and what he found from the previous studies 

i suggest authors to add section to discuss blockchain for privacy 

Author Response

Dear Reviewer

We thank you for your insightful comments and constructive suggestions, which helped us significantly improve the manuscript. For clarity, we have uploaded a copy of the original manuscript with all changes highlighted. Our point-by-point response to the reviewers' comments is attached to this letter. The comments are reproduced, and our answers follow in a different color (red) immediately after. We would also like to express our gratitude for allowing us to resubmit a revised copy of the manuscript.

Reviewer 4

the authors try to investigate the privacy needs of an IoT eco- 22 system as specified by industry standards.

  1. Motivation is not clear

A1- Thank you for your feedback. The motivation and contributions of the paper are highlighted in the revised version of the introduction section.

  1. Authors should discuss Industry 4.0 with more details and applications such as Industry 4.0 towards Forestry 4.0: Fire detection use case, Computing in the sky: A survey on intelligent ubiquitous computing for uav-assisted 6g networks and industry 4.0/5.0,  Digital twins collaboration for automatic erratic operational data detection in industry 4.0, Shape memory alloy-based wearables: a review, and conceptual frameworks on HCI and HRI in Industry 4.0, Autonomous Multi-Robot Collaboration in Virtual Environments to Perform Tasks in Industry 4.0.

A2- Thank you for your remarks. Industry 4.0 is a concept that involves the digitalization and automation of manufacturing processes, with a focus on creating intelligent factories that can adapt to changing demands and optimize production. While the paper does not explicitly discuss Industry 4.0, it focuses on the Industrial Internet of Things (IIoT), a critical component of Industry 4.0. IIoT is a crucial enabling technology for Industry 4.0, as it provides the necessary infrastructure for connecting devices, sensors, and machines in a factory setting. The applications mentioned in the question are all relevant to Industry 4.0 and demonstrate the diverse range of use cases that IIoT can enable. For example, using UAVs for fire detection in forestry represents a significant advancement in environmental monitoring and management. At the same time, wearables based on shape memory alloys offer new possibilities for monitoring worker health and safety in a factory setting. The concept of digital twins is also relevant to Industry 4.0. It involves creating virtual replicas of physical assets and processes, which can be used for simulation, optimization, and predictive maintenance. The use of multiple autonomous robots in virtual environments to perform tasks in Industry 4.0 represents another exciting application of IIoT, as it allows for the creation of flexible, adaptable manufacturing systems. Overall, the applications mentioned in the question demonstrate the broad scope and potential of IIoT in enabling the digitalization and automation of various industries, including forestry, manufacturing, and healthcare. As such, they represent essential areas for further research and development in Industry 4.0. Detailed explanations were added in the conclusion section.  

  1. at the end of every section, authors should summaries the finding and what he found from the previous studies

A3- Thank you for this insightful comment allowing us to clarify things further. At the end of each section, we summarize the findings from the mentioned studies.

  1. i suggest authors to add section to discuss blockchain for privacy

A4- That is a valuable suggestion. Blockchain is a distributed ledger technology that can provide enhanced security and privacy in the context of IIoT. Using blockchain, IIoT systems can create a decentralized, tamper-proof record of transactions and data, which can help ensure the integrity and confidentiality of sensitive information. Several potential use cases for blockchain in IIoT include secure data sharing between different organizations, secure and private communication between devices, and supply chain management. For example, blockchain can be used to create secure, auditable records of transactions in a supply chain, which can help to ensure the authenticity and quality of products. In terms of privacy, blockchain can enable secure and private data sharing between different organizations or individuals without needing a central authority to manage access control. This can be particularly important in industries that require strict privacy and security controls, such as healthcare or finance. Therefore, the main future direction of this research study (as was added in the revised conclusion) is a review of blockchain for privacy in the context of IIoT. It will provide insights into the potential benefits and challenges of using blockchain for privacy in IIoT implementations. It would also help highlight the growing importance of blockchain for enhancing security and privacy in the digital age.

Round 2

Reviewer 1 Report

The authors responded to most of my comments, but there is still room for improvement.

1. I am still missing some pictures/illustrations in the article.
2. Honestly, from a reader's point of view, it discourages me from reading an article that is just plain text without images.
3. Try adding some images to the article, such as an article organization chart.
4. Or IIoT architecture...something that would interest the reader.
5. In the article, I am missing a chapter in which you discuss what security risks result from the deployment of IIoT devices.
6.I suggest increasing the number of studies and adding a new discussion there to show the benefit. How IoT devices are vulnerable to common cyber attacks (DDoS and DRDoS).
7.Mitigation against DDoS Attacks on an IoT-Based Production Line Using Machine Learning
8.The vulnerability of securing IoT production lines and their network components in the Industry 4.0 concept
9.Smart Thermostat as a Part of IoT Attack
10.Industry Communication Based on TCP/IP Protocol
11.It would be a good idea to create a Discussion chapter in which you would evaluate your work and the sources used.
12.I think the Conclusion is unnecessarily long, some information from it could be moved to the Discussion chapter that I suggested you create in point 11.

Author Response

Dear Reviewer

We want to thank the reviewer for their insightful comments and constructive suggestions, which helped us significantly improve the manuscript. For clarity, we have uploaded a copy of the original manuscript with all changes highlighted. Our point-by-point response to the reviewers' comments is attached to this letter. The comments are reproduced, and our answers follow in a different color (red) immediately after. We would also like to express our gratitude for allowing us to resubmit a revised copy of the manuscript.

Reviewer 1

The authors responded to most of my comments, but there is still room for improvement.

  1. I am still missing some pictures/illustrations in the article. 2. Honestly, from a reader's point of view, it discourages me from reading an article that is just plain text without images. 3. Try adding some images to the article, such as an article organization chart. 4. Or IIoT architecture...something that would interest the reader.

A.1-2-3-4 According to your suggestions, several pictures/illustrations were added to the revised version of the paper. The paper now reads much better, and the work presented has been enhanced to a level appropriate for this journal's readership and scientific standing. Thank you for taking the time to read this carefully.

  1. In the article, I am missing a chapter in which you discuss what security risks result from the deployment of IIoT devices.

A.5 All mentioned information is included in chapter 4. Privacy Threats in the IIoT. Thank you for taking the time to revise our paper.

  1. I suggest increasing the number of studies and adding a new discussion there to show the benefit. How IoT devices are vulnerable to common cyber attacks (DDoS and DRDoS). 7.Mitigation against DDoS Attacks on an IoT-Based Production Line Using Machine Learning. 8.The vulnerability of securing IoT production lines and their network components in the Industry 4.0 concept. 9.Smart Thermostat as a Part of IoT Attack. 10.Industry Communication Based on TCP/IP Protocol

A.6-7-8-9-10 The paper focuses on privacy concerns in the IIoT ecosystem, as opposed to the more common focus on security risks. This is important because while security is certainly a critical concern in the IIoT, privacy is also a significant issue that must be addressed. In particular, the paper focuses on the privacy dimensions specified by industry standards for protecting personal data, including collecting, storing, sharing, and processing personal data.

11.It would be a good idea to create a Discussion chapter in which you would evaluate your work and the sources used.

A.11 According to your suggestions, a Discussion chapter was added to the revised version of the paper. Thank you for this constructive comment.

12.I think the Conclusion is unnecessarily long, some information from it could be moved to the Discussion chapter that I suggested you create in point 11.

A.1-2-3-4 According to your suggestions, the conclusion section was revised. Thank you for this helpful comment.

Reviewer 2 Report

There is no novelty in this manuscript. Therefore, not suitable for journal publication.

Author Response

Dear Reviewer

We want to thank the reviewer for their insightful comments and constructive suggestions, which helped us significantly improve the manuscript. For clarity, we have uploaded a copy of the original manuscript with all changes highlighted. Our point-by-point response to the reviewers' comments is attached to this letter. The comments are reproduced, and our answers follow in a different color (red) immediately after. We would also like to express our gratitude for allowing us to resubmit a revised copy of the manuscript.

Reviewer 2

There is no novelty in this manuscript. Therefore, not suitable for journal publication.

Thank you for your feedback. The novelty of the approach is highlighted in the revised version of the conclusion section as follows “The paper's novelty lies in its specific focus on privacy concerns in the IIoT ecosystem, as opposed to the more common focus on security risks. This is important because while security is certainly a critical concern in the IIoT, privacy is also a significant issue that must be addressed. In particular, the paper focuses on the privacy dimensions specified by industry standards for protecting personal data, including collecting, storing, sharing, and processing personal data.

The paper also provides an overview of contemporary approaches and solutions for addressing privacy risks in the IIoT. For example, the use of encryption and anonymization techniques can help protect sensitive information from unauthorized access, while access controls and authorization mechanisms can restrict access to data to only authorized individuals or devices. Monitoring and surveillance technologies can also detect and respond to security threats in real-time.

Furthermore, the paper's emphasis on protecting personal data in the context of criminal investigations or the execution of criminal sanctions is another novel aspect. This is particularly important in industries subject to regulatory oversight, where personal data may be subject to strict data protection regulations. The paper highlights the need to ensure that the processing of personal data complies with these regulations and other applicable laws while still allowing for the prevention, investigation, detection, or prosecution of criminal acts or the execution of criminal sanctions.

Overall, the paper's novel focus on privacy in the IoT ecosystem and its recommendations for addressing privacy risks make it valuable to the existing literature on the subject. By addressing privacy concerns and security risks, the paper provides a more comprehensive approach to ensuring the safety and privacy of the IIoT ecosystem and its end-users.

The importance of the paper to policymakers and organizations lies in its insights into the privacy needs and risks of the Industrial Internet of Things (IIoT) ecosystem. As the IIoT continues to transform the way organizations operate and the economy, policymakers and organizations need to understand and address the privacy concerns associated with this technology.

Also, from a policy perspective, the paper can help inform the development of regulations and guidelines that address privacy issues in the IIoT ecosystem. For example, by identifying the privacy dimensions specified by industry standards for protecting personal data, policymakers can develop regulations that require organizations to comply with these standards. Additionally, the paper's recommendations for addressing privacy risks can be used to inform the development of guidelines for organizations to follow to ensure the safety and privacy of their operations and end-users.

For organizations, the paper provides valuable insights into the privacy risks associated with IIoT and recommendations for addressing those risks. By implementing the strategies outlined in the paper, organizations can better protect sensitive information and ensure compliance with data protection regulations and other applicable laws. This can help organizations maintain the trust of their stakeholders and protect their reputation while benefiting from the efficiency and productivity gains associated with the IIoT.

In summary, the paper's insights into the privacy needs and risks of the IIoT eco-system are important for policymakers and organizations to understand to develop regulations, guidelines, and strategies that ensure the safety and privacy of the technology and its end-users.”

Reviewer 3 Report

Thanks to the authors for improving the manuscript. However, from the reviewer's point of view, Visualization is an important aspect of the publications.

Overall, the work contributes to the body of knowledge.

Author Response

Dear Reviewer

We want to thank the reviewer for their insightful comments and constructive suggestions, which helped us significantly improve the manuscript. For clarity, we have uploaded a copy of the original manuscript with all changes highlighted. Our point-by-point response to the reviewers' comments is attached to this letter. The comments are reproduced, and our answers follow in a different color (red) immediately after. We would also like to express our gratitude for allowing us to resubmit a revised copy of the manuscript.

Reviewer 3

Thanks to the authors for improving the manuscript. However, from the reviewer's point of view, Visualization is an important aspect of the publications. Overall, the work contributes to the body of knowledge.

A- Thank you for your remarks. According to your suggestions, several visualizations were added to the revised version of the paper.

Reviewer 4 Report

Authors address some comments but they didnot discuss with details how and why blockchain is important for  improving privacy. 

Author Response

Dear Reviewer

We thank you for your insightful comment and constructive suggestion, which helped us significantly improve the manuscript. For clarity, we have uploaded a copy of the original manuscript with all changes highlighted. Our point-by-point response to the reviewers' comments is attached to this letter. The comments are reproduced, and our answers follow in a different color (red) immediately after. We would also like to express our gratitude for allowing us to resubmit a revised copy of the manuscript.

Reviewer 4

Authors address some comments but they didnot discuss with details how and why blockchain is important for  improving privacy. 

A- Thank you for your feedback. The importance of improving privacy with blockchain technology is highlighted in the revised version of the 5. Privacy Requirements and Suggestions section as follows “Blockchain technology has the potential to significantly improve privacy by providing a decentralized, secure, and transparent way to store and transfer data. This is accomplished through several key features of blockchain, including encryption, decentralization, and immutability.

Encryption: One of the fundamental features of blockchain is encryption, which ensures that data is stored securely and privately. In a blockchain network, data is encrypted and can only be accessed by those with the required private keys. This makes it virtually impossible for unauthorized users to access or modify the data.

Decentralization: Blockchain is a decentralized technology, meaning no central authority controls the data. This makes it difficult for any single entity to access or manipulate the data. Instead, the data is stored across a distributed network of nodes, each with a copy of the blockchain ledger. This ensures that there is no single point of failure and that the data is highly resistant to tampering.

Immutability: The data stored on a blockchain is immutable, meaning that it cannot be modified or deleted once it has been added to the network. This is because each block in the blockchain is linked to the previous block, creating a continuous chain of data that is highly resistant to tampering. This ensures that the data is highly secure and trustworthy.

Taken together, these features make blockchain an ideal technology for improving privacy. By using blockchain to store and transfer data, users can be confident that their information is highly secure, transparent, and private. This is especially important in finance, healthcare, and e-commerce, where data privacy and security are critical concerns.

Moreover, the emergence of blockchain-based advanced privacy applications has further extended the privacy features of blockchain technology. These coins utilize advanced cryptographic techniques such as zero-knowledge proofs and ring signatures to ensure that the transactions are highly private and untraceable. This means that users can conduct transactions without revealing their identity, making them highly useful for privacy-conscious individuals and businesses.

In conclusion, blockchain technology is highly important for improving privacy due to its encryption, decentralization, and immutability features. These features ensure that data stored on the blockchain is highly secure, transparent, and resistant to tampering. As blockchain continues to evolve and mature, we will likely see even more advanced privacy features and applications emerge.

Round 3

Reviewer 1 Report

Even after two rounds of reviews, the authors did not manage to edit the article so that it was worthy of publication in the journal.

1. Added images are to be exclusively the work of the authors.

2. From the captions of the added images, we can see that they are reused from references: https://www.arcweb.com/, https://www.trendmicro.com/, https://www.rambus.com/, https://www.trendmicro.com/ https://dataprivacymanager.net/

3. If these images are reused for further publication, the author must obtain permission from the links provided.

Author Response

Dear Reviewer

We thank you for your insightful comment and constructive suggestion, which helped us significantly improve the manuscript. For clarity, we have uploaded a copy of the original manuscript with all changes highlighted. Our point-by-point response to the reviewers' comments is attached to this letter. The comments are reproduced, and our answers follow in a different color (red) immediately after. We would also like to express our gratitude for allowing us to resubmit a revised copy of the manuscript.

Reviewer 1

Comments and Suggestions for Authors

Even after two rounds of reviews, the authors did not manage to edit the article so that it was worthy of publication in the journal.

  1. Added images are to be exclusively the work of the authors.
  2. From the captions of the added images, we can see that they are reused from references: https://www.arcweb.com/, https://www.trendmicro.com/, https://www.rambus.com/, https://www.trendmicro.com/ https://dataprivacymanager.net/
  3. If these images are reused for further publication, the author must obtain permission from the links provided.

The figures in the revised version of the paper are exclusively from the authors' work. Thank you for this comment. 

Reviewer 4 Report

.

Author Response

Dear Reviewer

We thank you for your insightful comment and constructive suggestion, which helped us significantly improve the manuscript. For clarity, we have uploaded a copy of the original manuscript with all changes highlighted. Our point-by-point response to the reviewers' comments is attached to this letter. The comments are reproduced, and our answers follow in a different color (red) immediately after. We would also like to express our gratitude for allowing us to resubmit a revised copy of the manuscript.

Reviewer 4

Extensive editing of English language and style required

We reorganized the entire paper based on the reviewer's suggestions. In addition, typos and grammar errors have been corrected, and the English language used throughout the manuscript has been improved. The paper now reads much better, and the work presented has been enhanced to a level appropriate for this journal's readership and scientific standing. Thank you for taking the time to read this carefully.